# Full crystallographic orientation (c- and a-axes) of warm, coarse-grained ice in a shear dominated setting: a case study, Storglaciären, Sweden

Morgan E. Monz[1], Peter J. Hudleston[1], David J. Prior[2], Zachary Michels[1], Sheng Fan[2], Marianne Negrini[2], Pat J. Langhorne[2] and Chao Qi[3]

[1]Department of Earth and Environmental Sciences, University of Minnesota, Minneapolis, Minnesota, USA
[2]Department of Geology, University of Otago, Dunedin, New Zealand
[3]Key Laboratory of Earth and Planetary Physics, Chinese Academy of Sciences, Beijing, China

*Correspondence to:* Morgan E. Monz (monzx001@umn.edu)

**Abstract.** Microstructures provide key insights into understanding the mechanical behavior of ice.
Crystallographic preferred orientation (CPO) develops during plastic deformation as ice deforms dominantly by dislocation glide on the basal plane, modified and often intensified by dynamic recrystallization. CPO patterns in fine-grained ice have been relatively well characterized and understood in experiments and nature, whereas CPO patterns in "warm" (T > -10ºC), coarse-grained, natural ice remain enigmatic. Previous microstructural studies of coarse-grained ice have been limited to c-axis orientations using light optical measurements. We
present the first study of a-axes as well as c-axes in such ice by application of cryo - electron backscatter diffraction (EBSD) -and do so in a shear dominated setting. We have done this by developing a new sample preparation technique of constructing composite sections, to allow us to use EBSD to obtain a representative, bulk CPO on coarse-grained ice. We draw attention to the well-known issue of interlocking grains of complex shape, and suggest that a grain sampling bias of large, branching crystals that appear multiple times as island
grains in thin section may result in the typical multiple maxima CPOs previously identified in warm, coarse-grained ice that has been subjected to prolonged shear. CPOs combined from multiple samples of highly sheared ice from Storglaciären provide a more comprehensive picture of the microstructure and yield a pronounced cluster of c-axes sub-normal to the shear plane and elongate or split in a plane normal to the shear direction, and a concomitant girdle of a-axes parallel to the shear plane with a maximum perpendicular to the shear direction.
This pattern compares well with patterns produced by sub-sampling data sets from ice sheared in laboratory experiments at high homologous temperatures up to strains of ~1.5. Shear strains in the margin of Storglaciären are much higher than those in experimental work. At much lower natural strain rates, dynamic recrystallization, particularly grain boundary migration, may have been more effective so that the CPO represents a small, final fraction of the shear history. A key result of this study is that multimaxima CPOs in coarse grained ice reported
in previous work may be due to limited sample size and a sampling bias related to the presence of island grains of a single host that appear several times in a thin section.

## 1 Introduction

Ice sheets and glaciers play crucial roles in Earth's climate system, and understanding their dynamic behavior is essential for a variety of predictive purposes, including making projections of glacier and ice sheet

discharge and sea level rise (e.g. Bindschadler et al., 2013; Faria et al., 2014b; Dutton et al., 2015; Golledge et al., 2015; Bamber et al., 2019). In addition, glacial ice is a monomineralic rock that deforms at high-homologous temperatures as ice flows, and glaciers represent natural tectonic systems that undergo the equivalent of regional

high-grade metamorphism under known driving forces (Hambrey and Milnes, 1977; Van der Veen and Whillans, 1994). Similar to rocks in active orogens, flowing glacial ice develops both structures and CPOs that reflect the conditions and kinematics of deformation. Studying the internal structure of glaciers on the crystal scale provides key insights into ice mechanics, and aids in the understanding of tectonic processes (Hambrey and Milnes, 1977; Hooke and Hudleston, 1978; Faria et al., 2014b; Wilson et al., 2014; Hudleston 2015).

Quantifying flow behavior of ice under natural conditions is essential for the accurate incorporation of glacier flow into climate models and for using ice as an analog for high temperature deformation of crustal and mantle rocks (Hambrey, 1997; Wilson 1981; Faria et al., 2014b; Wilson et al., 2014). Glaciers move by two gravity-driven processes: (1) frictional sliding (including deformation of underlying sediments) of the ice mass over the underlying rock surface (e.g. Flowers, 2010 and references therein), and (2) slow, continuous creep

(flow) within the ice mass itself (e.g. Glen, 1955; Alley, 1992; Budd and Jacka, 1989; Cuffey and Paterson, 2010). Creep is governed by thermally-dependent, micro-scale deformation processes, and therefore participates in important thermo-mechanical feedbacks in the Earth's cryosphere, atmosphere and oceans. This is especially important because of the highly non-linear dependence of strain rate on stress (Glen, 1955; Budd and Jacka, 1989; Bons et al., 2018).

Terrestrial glaciers, ice sheets and ice shelves comprise crystals of hexagonal ice (Ih, Fig. 1a; Pauling, 1935; Faria et al., 2014b). As ice deforms plastically during flow, anisotropy in the form of a crystallographic fabric or crystallographic preferred orientation (CPO) develops due to a dominance of intracrystalline glide on the basal plane, and this is modified by recrystallization (Weertman, 1983; Duval et al., 1983; Faria et al., 2014b). Similar to other crystalline materials, such as rocks (e.g. Wenk and Christie, 1991), CPO development modifies the

internal flow strength (e.g. Steinemann, 1958; Lile, 1978; Pimienta and Duval, 1987; Alley, 1988; Alley, 1992; Azuma and Azuma, 1996; Gagliardini, 2009) and thus documenting natural ice CPOs provides insight into the large-scale flow rates of glaciers and ice sheets (e.g. Azuma, 1995; Azuma and Azuma, 1996; Faria et al., 2014b; Montagnat et al., 2014; Llorens et al., 2016a; Vaughan et al., 2017). The CPO of ice is commonly represented by the preferred orientation of c-axes. This is useful because the c-axis of an ice crystal is normal to

the basal plane (Fig. 1a), and glide on this plane dominates deformation (Duval et al., 1983). However, the orientations of a-axes are needed to fully characterize the orientation of ice crystals, and to better understand deformation mechanisms, since slip in the basal plane is not necessarily isotropic (Kamb, 1961), as has been demonstrated in recent shear experiments that result in the alignment of the a-axes (Qi et al., 2019; Journaux et al., 2019).

Coarse-grained (highly variable, but typically >20mm; see figure 2) ice is common at the base of ice sheets and in warm (T > -10ºC) glaciers. Work on coarse-grained ice is especially important because basal ice in ice sheets may accommodate much more of the ice flow than the colder ice higher up the ice column (e.g. Rignot and Mouginot, 2012; MacGregor et al., 2016), and clearly coarse-grained ice experiences large strains in valley glaciers (e.g., Kamb, 1959). Previous studies on coarse-grained ice have likely only measured partial CPOs,

typically by optical methods (c-axes only), and identified what may be apparent multimaxima patterns defined by isolated clusters of c-axes (Fig. 1b; e.g. Rigsby, 1951; Kamb, 1959; Jonsson, 1970). However, these

multimaxima patterns are incompletely understood and defined, in part because there has been no practical method for measuring the a-axes associated with such patterns. Measuring the a-axes means that we can tell whether two grains (in a 2D slice) with the same c-axis orientation also have the same a-axes and may be two slices through the same grain in 3D. Work on coarse-grained ice has been limited because methods used to measure CPOs are restricted to section sizes of 100mm x 100mm or smaller, which results in there being an insufficient number of grains needed to clearly define the CPO pattern without making use of multiple sections from a given volume of ice (Bader, 1951; Rigsby, 1968).

We aim to (1) better quantify the CPO patterns (c- and a-axes) associated with warm, coarse-grained ice using cryo-electron backscatter diffraction (cryo-EBSD), (2) understand how and why the apparent multimaxima CPO patterns develop, and (3) interrogate the relationships among multimaxima CPO patterns and local deformation conditions in the ice. To address these objectives, we combine results from fieldwork and laboratory analyses on Storglaciären, a small valley glacier in northern Sweden, and compare the results with the results of experimental work on ice deformation. Fieldwork included detailed mapping of structural features to provide a large-scale kinematic framework for our lab-based, microstructural study. Importantly, in the lab we developed a new sample preparation method to allow us to measure a representative volume and number of grains necessary for robust CPO characterization in coarse-grained ice using cryo-EBSD.

**2 Previous work**

Much of the pre-existing research on CPO development in natural ice has been done on ice cores from Antarctica and Greenland, and this has been nicely summarized by Faria et al. (2014a). Schytt (1958) produced the first microstructural study of deep polar ice from the ice core extracted from the Norwegian-British-Swedish-Antarctic Expedition of 1949-1952. Many studies of ice cores have been subsequently undertaken, on both Antarctica (Gow and Williamson, 1976; Lipenkov et al., 1989; EPICA community members, 2004; Seddik et al., 2008; Durand et al., 2009; Weikusat et al., 2009b; Azuma et al., 1999, 2000; Weikusat et al., 2017) and Greenland (Herron and Langway, 1982; Herron et al., 1985; Langway et al., 1988; Thorsteinsson, 1997; Gow et al., 1997; Wang et al., 2002; Svensson et al., 2003b; Montagnat et al., 2014). Studying microstructures in ice sheets offers the advantages of examining an extensive record of ice deforming under relatively simple kinematic conditions. As a result, CPOs in ice caps have been well defined and interpreted from ice cores, except perhaps at the base of ice sheets.

There are two typical end member c-axis CPO patterns that have been identified in experimental work, and these are useful in interpreting natural CPOs. At warm temperatures and lower strain rates, under uniaxial compression, the c-axes define an open cone shape or small circle girdle at 30-60° about the axis of compression on a CPO plot (Fig. 1c; e.g. Jacka and Maccagnan, 1984; Alley, 1988; Budd and Jacka, 1989; Jacka and Jun, 2000; Treverrow et al., 2012; Piazolo et al., 2013; Montagnat et al., 2015; Vaughan et al., 2017; Qi et al., 2017). Whether this CPO occurs in nature is less clear. Possible examples are described at the center of ice domes, where they would be expected (e.g. Hooke and Hudleston, 1981; Lile et al., 1984; Gow and Meese, 2007). There are certainly fabrics close to open cones (sometimes referred to as small circle girdles) in the upper parts of many polar ice cores (e.g. Ross ice shelf, Gow and Williamson, 1976; Byrd Station, Gow and Williamson, 1976; Camp Century; Herron and Langway, 1982; Cape Folger, Thwaites et al., 1984; Dye 3, Herron et al.,

1985; Siple Dome, DiPrinzio et al., 2005; Siple Dome, Gow and Meese, 2007; NEEM, Montagnat et al., 2014). Additionally, some CPOs in coarse-grained ice at the base of ice sheets have been identified as possible open cones or modifications of open cones (e.g. Byrd Station, Gow and Williamson, 1976; Tison et al., 1994; GRIP,

Thorsteinsson et al., 1997; GISP2, Gow et al., 1997; Siple Dome, DiPrinzio et al., 2005; Siple Dome, Gow and Meese, 2007), even though these types of fabrics typically show clustering that is interpreted as a multimaxima CPO. It is important to note, however, that the eigenvalue technique of fabric representation, often used with more recent analyses, does not distinguish between small circle girdles and multimaxima fabrics (Fitzpatrick et al., 2014), and is inappropriate for multimaxima fabrics.

Under simple shear conditions, the basal planes of ice crystals dominantly align with the shear plane, and the c-axes form an asymmetric bimodal distribution with both a strong maximum perpendicular to the shear plane and a weaker secondary cluster offset at an angle antithetic to the rotation associated with the shear direction (Fig. 1c). The angle between the two clusters varies with shear strain, and the weaker cluster ultimately disappears with increasing strain leaving a strong single maximum pattern normal to the shear plane (Fig. 1d;

e.g. Duval, 1981; Bouchez and Duval 1982; Budd and Jacka, 1989; Budd et al., 2013; Qi et al., 2019; Journaux et al., 2019). This dual maxima pattern of CPO development under simple shear has been described in nature (Hudleston, 1977a; Jackson and Kamb, 1997).  It is probable that the strong single vertical maximum seen in many ice cores from Antarctica and Greenland are associated with zones of sub-horizontal simple shear (e.g. Gow and Williamson, 1976; Azuma and Higashi, 1985; Paterson, 1991; Alley, 1992; Tison et al., 1994;

Thorsteinsson et al., 1997; Faria et al., 2014a; Montagnat et al., 2014). However, there are almost no new data for the evolution of CPO of natural ice in shear zones, because there is very little close control of strain gradients in natural ice.  Nearly all the published data comes from laboratory experiments.  As far as we are aware there is still only one study of fabrics in natural ice constrained to be from a well-defined shear zone (Hudleston, 1977).

An enigmatic CPO pattern can develop in valley glaciers and deep in ice sheets in coarser grained ice that has undergone significant recrystallization. This pattern is always associated with warmer (T > −10ºC) conditions and an increase in grain size, and is characterized by 3-4 maxima (sometimes with submaxima), arranged around an axis that is vertical in ice sheets (Gow and Williamson, 1976; Thwaites et al., 1984; Goossens et al., 2016), and perpendicular to foliation in valley glaciers (Fig. 1b, Fig. 2; Kamb, 1959; Allen,

1960; Budd, 1972; Jonsson, 1970). In most cases, given the coarse grain size (Fig. 2a), the number of grains measured per thin section is small, usually no more than ~100. This may or may not be enough to reveal a mechanically significant CPO pattern (Fig. 2b; Rigsby, 1960). By contrast, CPO plots produced for fine-grained ice and other deformed crystalline materials typically include data from several hundred unique grains/crystals, which can usually be collected from a single sample section. This would be difficult or impossible to

accomplish with coarse-grained ice.

        Previous studies of coarse-grained ice in valley glaciers done by Rigsby (1951) on Emmons glacier, Kamb, (1959) on Blue Glacier, and Jonsson (1970) on Isfallsglaciären used light optical measurements to delineate a CPO characterized by a multimaxima pattern of the type described above, but were limited to measuring c-axis orientations. Such studies used a Rigsby universal stage to individually orient c-axes (Langway, 1958), and they

demonstrated a relationship of the overall c-axis CPO to other structural elements, with the pole to foliation typically located centrally among the maxima (Kamb, 1959; Jonsson, 1970).

Possible analogues to the multimaxima CPOs found in nature have been produced in experiments by Steinemann (1958) and Duval (1981), in both cases at temperatures near the melting point and under torsion-compression conditions. The maxima developed at high angles to the shear plane. It should be noted however, 165 that the grain size in the experiments is much smaller than in natural ice with these CPOs.

Ice with the multi maxima CPO in valley glaciers (Rigsby, 1951; Meier et al., 1954; Kamb, 1959; Higashi, 1967; Jonsson, 1970; Fabre, 1973; Vallon et al., 1976; Tison and Hubbard, 2000; Hellmann et al., in review) and deep in ice sheets (Gow and Williamson, 1976; Matsuda and Wakahama, 1978; Russell-Head and Budd, 1979; Gow et al., 1997; Diprinzio et al., 2005; Gow and Meese, 2007; Montagnat, 2014; Fitzpatrick et al., 2017; Li et 170 al., 2017) is comprised of large, branched crystals that lack undulose extinction and have irregular, lobate grain boundaries (Fig. 2a; Fig. 3). Individual grains are so large that even with the maximum size thin section (using any method of analysis), the exact shape and extent of individual grains remain unknown. Additionally, the branching nature of these crystals may result in sectioning artifacts that lead to apparent "island grains"— branches of the same grain appearing multiple times throughout one 2D thin section (Fig. 3: e.g. as illustrated in 175 glacial ice by Bader (1951) and Rigsby (1968), in sea ice by Dempsey and Langhorne (2012), and in quartz (Stipp et al., 2010). Without a complete crystal orientation – one that includes ice a-axes – it is difficult to confirm the existence of such island grains and determine their effect of the characterization of a representative CPO. Early work tried to address the problem of sample size by making multiple sections from different parts of a sample or core, spacing thin sections between 5 and 15cm intervals, (Rigsby, 1951; Gow and Williamson, 180 1976; Thwaites et al., 1984) or taking them from more than one sample (Kamb, 1959). Nonetheless, there remains the uncertainty about whether the maxima are truly distinct or reflect repeated measurements of individual grains. It might be noted that in recent work little or no explicit attention is given to the problem of sample size in coarse-grained ice (see Dahl-Jensen et al., 2013; Montagnat et al., 2014; Fitzpatrick et al., 2014; Li et al., 2017), and to the significance of possible island grains on fabric (see Diprinzio et al., 2005; Gow and 185 Meese, 2007; Dahl-Jensen et al., 2013; Montagnat et al., 2014; Fitzpatrick et al., 2014; Li et al., 2017). This problem may not have been highlighted, as CPO in coarse-grained ice was not the sole focus of these ice core studies.

A number of interpretations have been proposed for the multimaxima CPOs, though it is clear that there is no single explanation that can be applied to all cases. Earlier studies made efforts to quantify an angular 190 relationship between clusters of c-axes, but no consistent relationship could be found, and a mechanism that produces such a pattern – with regular angular relationships or otherwise – has not been established. For one thing, the number, shape and relative intensity of the maxima that define the CPO are variable (e.g. Rigsby, 1951, 1960; Kizaki, 1969; Jonsson, 1970), even though the "ideal" shape is classified as rhomboid or diamond (Rigsby, 1951, 1960). It has been proposed that the multimaxima pattern may be the result of mechanical 195 twinning (Matsuda and Wakahama, 1978), although the texture in thin section gives little indication of this. (It should be noted that twinning can only be investigated if both a- and c-axes are known). It is often assumed that CPOs are related to the state of stress, and that the maxima reflect the basal plane alignment with orientations of high shear stress (Duval, 1981). If this were the case, there should be no distinction between CPOs formed in coaxial and non-coaxial kinematics, there should be just two maxima, and there should be a consistent 200 relationship between fabric elements and the principal stress directions. However, in pure shear, found in the center of the ablation zone near the surface of valley glaciers, where ice undergoes longitudinal compression, the

maximum principal stress is horizontal and the multimaxima pattern is centered about the axis of compression (Hellmann et al., in review), which in the case of the Blue Glacier is also the pole to foliation (Kamb, 1972, fig. 17b). By contrast, in simple shear, assumed to hold near glacier margins, the maximum principal stress is inclined at 45° to the foliation (shear plane), and the maxima are arranged about the normal to the foliation (Kamb, 1959) and not centered about the maximum principal stress direction.

A number of previous studies proposed that recrystallization dominated by grain boundary migration results in the multimaxima CPOs (Rigsby, 1955; Gow and Williamson, 1976; Gow et al., 1997; Duval, 2000; Diprinzio et al., 2005; Gow and Meese, 2007; Montagnat et al., 2014). While dynamic recrystallization likely plays an important role, these studies do not provide an interpretation as to why recrystallization results in the geometrically spaced clustering of c-axes rather than the well understood patterns found in fine-grained ice. Some authors suggest the multimaxima pattern illustrates the transition between small circle girdles and single maximum CPOs (e.g. Rigsby 1955; Gow and Williamson, 1976; Gow and Meese, 2007; Fitzpatrick, et al., 2017), but again do not provide a reason this would result in several distinct maxima.

We argue that previously employed methods have most probably not been able to clearly determine a representative CPO for glacial ice consisting of coarse, branching crystals. Optical studies using the Rigsby stage, such as those illustrated in figure 2, which accommodates 100mm x 100mm thin sections, are time consuming, especially when many sections must be made for one sample, and are limited not only by incomplete crystal orientations, but also by data resolution. Automatic ice texture analyzers (AITA), which can also accommodate larger grain sizes, use an image-analysis technique under cross-polarized light to determine c-axes (Russell-Head and Wilson, 2001; Wilen et al., 2003). AITA analyses are attractive for speed and data resolution, but are also limited by incomplete crystal orientations (Russell-Head and Wilson, 2001). For both the Rigsby stage and AITA methods, it is not possible to relate two grains with the same c-axis orientation in two-dimensions to the same parent grain, unless traced through an undetermined number of successive thin sections. This is near impossible for all grains since the exact size and shape of the crystals remains undefined.

Three methods: etching (Matsuda 1979; Matsuda and Wakahama 1978), semi-automated Laue diffraction (Miyamoto et al 2011; Weikusat et al 2011), and EBSD (Dingley, 1984; Prior et al., 1999) enable the measurement of full crystallographic orientations in ice (Obbard et al., 2006; Obbard and Baker, 2007; Weikusat et al., 2017; Kim et al., 2020). Etching is time intensive and the results are of low angular resolution. The other two methods produce results of high resolution. Laue X-ray diffraction has been applied as a spot based method while EBSD provides the orientation of every pixel measured.

Cryo-EBSD as a technique was first applied to ice in 2004 (Iliescu et al 2004), and modern cryo-EBSD methods enable routine work on water ice (Prior et al 2015). CPOs derived from EBSD datasets include a-axis orientations and provide a comprehensive view of ice microstructure that can improve our knowledge of the CPO and its relation to ice flow mechanisms on the grain scale. In addition, the speed, angular precision, and spatial resolution attainable with modern EBSD systems offer major advantages over optical methods. However, until now, EBSD has not been applied to warm, coarse-grained ice because a sample of maximum size for analysis (60mm x 40mm: Prior et al., 2015; Wongpan et al., 2018) will only contain a few grains. The procedure we apply in this paper addresses this limitation.

**3 Glaciological Setting**

Storglaciären is a small polythermal valley glacier located in the Tarfala Valley in northern Sweden (Fig. 4). The glacier is 3.2km long, extending in an E-W direction, with a total surface area of 3.1km$^2$. A cold surface layer (annual mean of -4.0ºC) (Hooke et al., 1983a; Holmlund and Eriksson, 1989; Pettersson et al., 2007) of variable thickness (20-60m) (Holmlund and Eriksson, 1989; Holmlund et al., 1996; Pettersson et al., 2003), and a cold-based margin and terminus (annual mean of -4.0ºC) (Holmlund et al., 1996; Pettersson, 2007), characterize the ablation zone (Holmlund et al., 1996b). The thermal regime influences glacier dynamics; the center of the glacier undergoes basal sliding, but the margins and terminus are frozen to the overlying and marginal rock (Holmlund et al., 1996), causing most of the deformation in these areas to be a result of creep (Pettersson et al., 2007). Storglaciären was chosen because: (1) a compilation of preexisting information on surface velocities and seasonal changes gathered over many years exists to provide background for the study; (2) the multimaxima pattern has been observed optically in strongly sheared marginal and basal ice (Fig. 2), and (3) because it is comparatively easy to access.

Primary stratification is easily identified above the equilibrium line on the glacier as gently undulating layers roughly parallel to the ice surface. The ice in Storglaciären undergoes horizontal compression and shortening as it enters the valley from the accumulation cirques, and this amplifies the slight undulations in primary stratification, causing upright, similar folds (Ramsay, 1967) near the margins of the valley (walls) where shearing, which combines with shortening, is most intense. Folds range from centimeter to meter amplitude, and generally have axial surfaces that are vertical near the margins and contain the flow direction. They are associated with an axial planar foliation and have hinges that plunge gently west, away from the flow direction. Foliation develops from pre-existing stratification, veins and fracture traces where shear is most intense (e.g. Hambrey, 1975; Roberson, 2008; Jennings et al., 2014), and is defined by variations in crystal size, shape and bubble concentration and distribution (e.g. Allen et al., 1960; Hambrey, 1975; Hambrey and Milnes, 1977; Hooke and Hudleston, 1978). Foliation tends to become perpendicular to the maximum shortening direction, and thus rotates with progressive shear towards parallelism with the flow direction along the glacier margins (Fig. 4; Ragan, 1969), reflecting cumulative strain (Hambrey and Milnes, 1977; Hooke and Hudleston, 1978; Hambrey et al., 1980; Hudleston, 2015).

## 4 Methods

### 4.1 Field Work

Detailed mapping in 2016 and 2018 on the surface of the glacier provides the structural framework for this study. Data collection was focused on multiple transects across the glacier in the ablation zone. Relevant data, presented in Figure 4, highlight the relationship of the structures to one another and the known kinematics.

We collected samples from eight areas of intense deformation in the ablation zone during the 2018 field season. For the purposes of this paper, we are focusing on three samples from the intensely sheared southern margin (SG23, SG27, and SG28) (Fig. 4) because they are from a small area with well-defined kinematics. The other samples collected in 2018 were spread out across the glacier in various and more complex local settings, and were not clustered in such a way that data could be combined for a strong interpretation, and thus do not

contribute to the arguments we present here. We excavated 10-20cm of surficial ice before sampling to avoid a layer of solar-damaged, recrystallized ice. Damaged ice was broken up using an ice axe and removed with a shovel. Blocks of ice were removed from the glacier using a small chainsaw. Each sample was ~15x15x30cm,

oriented such that the top of the block was parallel to the glacier surface, and the long axis was N-S, perpendicular to the flow direction. The shear plane, used to define the kinematic reference frame for subsequent microstructural analyses, is assumed to be parallel to the foliation. Samples were immediately shaded with a tarp upon removal to avoid solar damage, then labeled and insulated with ice and jackets to be transported off the glacier. We trimmed samples with a band saw in a cold room at the University of Stockholm, Sweden, and

marked the top north edge with a notch. We transported these samples to the University of Otago, New Zealand, in doubly insulated Coleman Xtreme 48L wheeled coolers, each of which can only contain four samples, to be stored in a biohazard freezer set to -31ºC. Samples remained below -20ºC for the entire transport pathway.

**4.2 Sample preparation**


We prepared samples for EBSD mapping and microstructural analysis in a cold room (-20ºC) at the University of Otago. To do this, we developed a novel composite sample preparation method to maximize the number of grains collected and minimize the number of repeated grains, in order to obtain a representative CPO. We made at least two composite sections for imaging from each of the eight samples, totaling 18 composite

sections. We emphasize that we are not the first to combine orientation data from multiple oriented sections to overcome the problem of sampling when dealing with very large grain sizes (e.g. Rigsby, 1951; Kamb, 1959; Gow and Williamson, 1976; Thwaites et al., 1984). Our method provides a way of dealing with the specific technical challenges of using EBSD for coarse-grained ice since the time/ resource limitation for EBSD is time on the instrument and with fast EBSD speeds, the sample exchange rather than the analysis time becomes the

limit.  Making composite sections enables us to collect data equivalent to 10 to 20 full sample sections with only one exchange of samples, taking a half day of SEM time rather than what would otherwise be two weeks.

The sample preparation procedure is highlighted in figure 5. We initially cut each sample block into three 5cm thick slabs perpendicular to the foliation. We then divided each slab into rods, spaced by 5cm, perpendicular to the flow direction and to the foliation.  These rods were cut such that they were staggered

between sequential slabs, and a series of ~2mm thick slices were cut off of the bottom or top of each rod (easiest to divide each rod into equally spaced cubes before cutting slices due to the delicacy of individual slices). Each slice was labeled, oriented, and stacked sequentially between two wooden blocks within a clamp to hold loose slices together before being cemented. We wrapped wet paper towels around the compiled stack to adhere the slices into a coherent block, ~3.6x5x5 cm. We then cut these blocks in half to generate a flat composite surface,

labeled each half, and returned one to storage for future use. We mounted sections on 4x6cm copper and aluminum ingots in the cold room using the freeze-on technique outlined by Craw et al. (2018) and, to ensure secureness, used thin slices of wet paper towels around the edges in contact with the ingot. The exposed surface was then flattened and polished using progressively finer sand paper and then cooled slowly to ~ -90ºC before being inserted into the SEM.

We note that there are associated errors of misorientation with each step. We consider the process in several stages. Each sample is first squared into a rectangular prism, with one side vertical and another parallel to

foliation, using guides to ensure perpendicularity. Guides are then used for each of steps 1-4 (Fig. 5), cutting the sample progressively into slabs, rods, cubes and slices. The errors involved in each stage of this process are estimated to be less than 0.5°. The error involved in slight twisting between slices during assembly into a composite section is estimated to be no more than 1°. Combining data from two or three composite sections in a sample adds only possible errors of misalignment in mounting for EBSD measurement. This is estimated to be no more than 0.5°. These sum to give possible errors of misorientation of the slices making up the composites and thus of the pole figures derived from them of 3-4°.

Whole sections of certain areas of the original blocks were prepared for examination, to mitigate loss of information on internal structure due to the small slices for the composite sections. Slabs cut perpendicular to foliation (first step in composite preparation) were polished using progressively finer sandpaper and allowed to sublimate overnight, then illuminated using low angle light, which revealed grains intersecting the surface. Areas of interest in these slabs were targeted for whole section analysis. At least two whole sections were taken from each sample.

It is important to note that the copper and aluminum ingots on which the samples were mounted were up to 40 x 60mm because that is the maximum size the SEM can analyze without significant risk of sample crashes (Prior et al 2015 show a larger sample but 40mm x 60mm is now the standard max size). This size pushes the limits of the instrument, and therefore we aimed to make sections that were not quite 60mm wide. We experimented with the width of the composite slices, initially starting with 5mm (see Fig. 7, SG23 composite 2 EBSD image—this was the first composite constructed), and determined that in order to maximize the number of grains, we needed to use more slices that were thinner. We ultimately aimed for 36 spaced slices per sample - 18 per composite - that were each approximately 2mm wide. This allowed extra room, which was important because different bubble concentrations throughout the sample made certain areas more fragile than others. Slices in areas with a high bubble concentration needed to be a bit wider (2.5-4mm). Ultimately, most of the composite sections were between 36mm and 50mm wide. Thus it was practical considerations that limited the width of the sections we produced. Additionally, for whole sections, we were interested in examining the internal structure of the largest grains, which included subgrain boundaries, and also the misorientations between grain boundaries. Many of the sections measured were mounted on the larger ingots (40mm x 60 mm), but due to the limited number of these, some were mounted on smaller ingots (30mm x 30 mm). All produced similar analytical results.

### 4.3 Orientation data collection

A Zeiss Sigma variable pressure field-emission-gun Scanning Electron Microscope (SEM) fitted with a Nordlys EBSD camera from Oxford Instruments was used for EBSD analyses. The instrument is fitted with a custom-built cryo-stage that is continuously cooled by liquid nitrogen from an external dewar via a copper braid connection (Prior et al., 2015). The stage is cooled below -100ºC prior to sample insertion. During the transfer process, the sample did not exceed -80ºC. Once the stage cooled back down to -100ºC, we vented the SEM chamber, allowing the stage temperature to rise to -75ºC, inducing a sublimation cycle outlined by Prior et al. (2015) to remove any residual frost from the sample surface before imaging.

We collected full cross-sectional orientation maps of whole sections (e.g. Fig. 6a,b) and composite sections (e.g. Fig. 7a) at a 50μm step size in order to balance data resolution with such a coarse grain size. SEM settings for EBSD acquisition were a stage temperature of ~-90ºC, a chamber pressure of 3-5Pa, an accelerating voltage of 30kV, a beam current of ~60-70nA, and a sample tilt of 70º. Each large section takes >1 hour to analyze at this coarse step size, additional time to analyze any areas of interest in finer detail, and another hour to do a sample exchange, run the sublimation cycle to clean frost off of the sample for imaging, bring the stage down to the correct temperature, and set up another analysis. When all goes smoothly, only 3-4 sections can be analyzed per day.

EBSD data were collected using the Aztec Software from Oxford Instruments and exported into Oxford-HKL Channel 5. We used EBSDinterp 1.0, a graphic user interface based MATLAB® program developed by Pearce (2015) to reduce noise and interpolate non-indexed EBSD data points using band contrast variations. Noise reduced data were then processed using MTEX, a texture analysis toolbox for MATLAB® (Bachmann et al., 2010), to determine full crystallographic orientations, intergranular misorientations, grain boundaries and to calculate one-point-per-grain CPO plots (Mainprice et al., 2015). The overall CPO in our samples is best represented using one-point-per-grain plots rather than all-pixel orientation plots due to the area bias introduced by larger grains in a small sample size. We note that representing the data using all-pixel orientations does take into account the issue of parent grains with satellite island grains, but only if the sample is large enough to contain a sufficient number of grains to provide a truly representative CPO (Appendix A). If the sample does not contain a representative number of grains, as is often the case with coarse-grained ice, then using one-point-per-grain provides a more representative CPO (Fig. A1). The kinematic reference frame used for plotting CPO is shown in figure 4.

**5 Results**

**5.1 Field Work**

Orientation measurements of bedding and foliation are consistent with previous observations on Storglaciären and other valley glaciers. Bedding is difficult to distinguish from foliation at the margins of Storglaciären, but more obviously recognizable in the center of the glacier. Although locally variable due to folding, in the center of the ablation zone, bedding generally dips shallowly west. Along the margins, the foliation is subvertical, dipping steeply inwards towards the center of the glacier (Fig. 4). In the center towards the front of the glacier, the foliation becomes progressively shallower and dips shallowly up glacier where sheared basal ice is closer to the surface (Fig. 4). The combination of transformed stratification and foliation in the ablation zone forms a series of arcs on the surface reflecting in three dimensions an overall nested spoon arrangement, opening up glacier, much as described by Kamb (1959) for the Blue Glacier.

**5.2 Microstructure**

Grains are locally variable in size, ranging from 1mm to >90mm. They have no apparent consistent shape preferred orientation (SPO). Air bubbles exist as a secondary phase and are found both within grains and on

grain boundaries (Figs. 2a and 6a,b). Broadly, there is an inverse correlation between bubble concentration and grain size, and also between bubble concentration and grain boundary smoothness.

### 5.2.1 Whole Section

The size of an individual whole-section is determined by the technique used for the analysis. For U-stage work it is 100mm x 100mm, whereas for EBSD work it is 40mm x 60mm. Neither section size is large enough to clearly measure the coarse crystal size, but such sections capture the complexity of grain boundaries and crystal shapes. Larger crystals have lobate-cuspate boundaries (Fig. 2a; Fig. 6a,b), and many grains are larger than the size of the thin section. Many larger grains within one measured section have the same color in thin section under cross-polarized light and are shown to have the same crystallographic orientations by EBSD data, with near identical c-axis and a-axis orientations (Fig. 2; Fig. 6,b,c). We chose to show sections from the smaller ingots (~30mm x 30mm) (Fig. 6a,b) because the data resolution was high (not many mis-indexed points/holes in the data, or cracks in the section) in comparison with those from the larger ingots. These sections highlight all the features we discuss.

Misorientation profiles A-A' (Fig. 6a) and B-B' (Fig. 6b) show that the orientation gradient across individual grains is low. The pixel-to-pixel scatter, mostly less than ±0.5º is typical of the angular error for fast EBSD acquisition (Prior et al., 1999). Profile A-A' shows an abrupt change of about 4º across a subgrain boundary, and no distortion within the grain or subgrain. In nine whole sections analyzed for this study, ~15% of grains contain subgrain boundaries, with misorientations ranging between 2.5º and 5.5º (e.g. Fig. 6a). Profile B-B' shows a grain that has no internal distortion, and profile C-C' shows an orientation change of about 2.5º across ~20mm. The statistics of misorientation between every pixel and the average orientation for that grain (Fig 6e) shows that 99% of these misorientations are below 2.5º. There is very little orientation spread, a measure of lattice distortion in the grains in this and all of the other sections shown.

### 5.2.2 Composites

Several c-axis maxima clustered around the normal to the shear plane are present in individual samples and this is largely independent of whether we plot all measured pixel orientations or one-point-per-grain orientations (Fig. 7b,c,d). The maxima in the all-pixel diagrams (Fig. 7b) have different relative intensities compared to those in the one-point-per grain CPO plots (Fig. 7c,d), reflecting the increased weight given to the larger grains in the per pixel data. In either case, many c-axes within an individual cluster are only separated by 3º-5º. The a-axes define a diffuse girdle, parallel to sub-parallel with the shear plane, containing three distinct clusters (Fig. 7e). Each cluster is elongate towards the pole to foliation.

When composites SG23, SG27 and SG28, which are in the same kinematic reference frame, are individually plotted as one point per grain, and these results are combined on one CPO plot, the multimaxima nature of the pattern diminishes (Fig. 8). The composite pattern has one c-axis maximum roughly perpendicular to the shear plane, that is elongated or split into two maxima aligned in a plane normal to the shear direction, and an a-axis girdle parallel with the shear plane with a concentration of a-axes perpendicular to the shear direction (parallel to the inferred vorticity axis of flow). Two weak c-axis sub-maxima are offset from the main

maximum in a plane perpendicular to the vorticity axis: the more distinct one ~30º synthetic to the shear direction and the less distinct one ~50º antithetic to the shear direction (Fig. 8).

It is important to note that another source of error in creating Fig. 8 results from combining data from the three samples on to one pole diagram. The reference frame for this is the foliation plane (xy-plane with vertical, x, recorded on each block when removed from the glacier.) The error in combining data from the three samples is estimated to be no more than 1°. Adding this source of error to those associated with sample preparation (see above) we estimate the uncertainties in positioning points on the pole diagrams Fig. 8 to be no more than 6°. The overall effects of such errors are likely to modestly diffuse rather than strengthen the maxima shown, but they will not modify the basic pattern. We assert that the measurements we have made are sufficient to establish the main features of the fabric in Fig. 8.

## 6 Discussion

### 6.1 Whole sections

EBSD maps of whole sections confirm that island grains are likely part of the same larger grain based on identical full crystallographic orientations (Fig. 6a,b). Individual grains within a two-dimensional surface that have exactly the same orientation or a slight misorientation are likely branching segments of the same grain, or subgrains of the larger grain in three dimensions (Fig. 3; Fig. 6b,c). Even small (30mm x 50mm) 2D sections can contain 3-5 island grains that have the same orientation (Fig. 6b,c). By appearing several times in the same section, some of the larger crystals amplify individual maxima within the overall CPO pattern typically identified in warm, coarse-grained ice. This may particularly be the case in studies that only use ~100 or fewer grains to identify a c-axis pattern, because if 10-15 islands comprising the same grain were measured as separate grains, that would automatically lead to a c-axis maximum due to that grain.

Whole section analyses also allowed us to better understand the deformation mechanisms. While some subgrains are present in the suite of whole sections analyzed, most crystals show little evidence of significant lattice distortion. Individual grains are relatively strain free (Fig. 6e). A lack of intragranular distortion, combined with the presence of lobate-cuspate grain boundaries, no visible shape preferred orientation, and evidence of grain boundary drag around bubbles (e.g. Fig. 6a), similar to pinning effects discussed by Evans et al. (2001), suggests that recrystallization in these samples is dominated by grain boundary migration (Urai et al., 1986). These interpretations are consistent with those in microstructural studies of experimentally deformed ice at high temperatures (e.g. Kamb, 1972; Montagnat et al., 2015; Vaughan et al., 2017; Journaux et al., 2019), and natural ice samples deformed at relatively high temperature (Duval and Castelnau, 1995).

### 6.2 Composite sections and combined samples

c-axis patterns for *individual* samples appear to represent typical multimaxima CPO patterns of the kind that have previously been identified in warm, coarse-grained ice (Fig. 7b,c), with 2-3 strong maxima and 1-2 weaker maxima all centered about the pole to foliation. However, on CPO plots of one-point-per-grain c-axes, we interpret the small angular difference between many of the individual points as most likely due to branched

grains appearing multiple times throughout the sample section and thus being counted more than once, consistent with observations made on whole sections. This interpretation is strengthened because c-axis clusters in Fig. 7 are coupled with corresponding a-axis clusters. The small 3º-5º misorientations of individual c-axes within a cluster are likely due to the combination of slight non-parallelism and rotation of slices that occurred during the sample preparation process (as described above) and the internal structure of individual grains. On this basis, we propose that multimaxima patterns such as those described in previous studies may be an *apparent* result caused by grain sampling bias, with some samples containing fewer than 30 *unique grains* within a set of 100 apparent grains (i.e. the case assuming no multiple counting). Thus, even for the composite samples, the data in Fig. 7 likely do not truly provide a representative one-point-per-grain CPO. Combining sections for SG23, SG27, and SG28 provides a more representative dataset (fig. 8), reducing but not entirely eliminating the bias.

### 6.3 Comparison with Experimental Results

Only two published sets of experiments document both c-axis and a-axis CPOs in simple shear in ice, and those are the ones by Qi et al. (2019) and Journaux et al. (2019). The results of these two sets of experimental studies are coherent, exhibiting two clusters of c-axes, a strong cluster normal to the imposed shear plane at all strains, and a secondary cluster in a profile plane antithetic to the imposed shear direction at lower strains. Both studies highlight the disappearance of the weaker maximum, and an enhancement of the stronger maximum with high shear strains. Except for grain size, we interpret microstructures in the ice from both the warm temperature experiments by Qi et al. (2019) and the experiments done by Journaux et al. (2019) to be similar to those in our samples from Storglaciären (including c- and a-axis CPOs) and to other examples (including only c-axis CPOs) of warm, natural ice (Rigsby, 1951; Kamb, 1959; Jonsson, 1971). Individual grains from these "warm" experiments by Qi et al. (2019) and Journaux et al. (2019) are characterized by ameboidal shapes and lobate boundaries, and portray little to no shape preferred orientation in the two dimensional plane.

We provide a more detailed comparison of our CPOs from natural ice to experimentally obtained CPOs from two warm temperature (-5ºC) direct shear experiments by Qi et al. (2019), at relatively low (γ=0.62) and high (γ=1.5) strains. A major advantage of using an experimental dataset for our comparison is that it comprises hundreds more grains than can be measured in a single sample of coarse-grained glacial ice – even with using the novel composite-section sampling techniques addressed in this paper. Given the similarity in grain-shape characteristics and deformation temperature, and owing to the greater number of analyzed crystal orientations, we argue that CPO patterns from both the Qi et al. (2019) and Journaux et al. (2019) samples represent an excellent analogue for crystallographic texture evolution of ice along the margins of Storglaciären.

Orientation data from Qi et al. (2019) show well-defined CPO patterns with a two-cluster c-axis pattern: a strong c-axis maximum perpendicular to the shear plane, and a c-axis sub-maximum rotated from the dominant maximum 45º-70º in a direction antithetic to the shear induced rotation (Fig. 9). The angle between the strong maximum and sub-maximum decreases with increasing shear strain. Clusters of c-axes are somewhat elongate in a plane normal to the shear direction.

The elongation of the main c-axis maximum in a plane normal to the shear plane and in a direction perpendicular to the shear direction is found both in simple shear experiments (Kamb, 1972; Bouchez and

Duval, 1982; Journaux et al., 2019; Qi et al., 2019) and in experiments involving simple shear with the added effect of compression or flattening normal to the flow plane (Kamb, 1972; Duval, 1981; Budd et al., 2013; Li et al., 2000). It is also found in our samples (Fig.8). There are many proposed explanations, a combination of which likely tells the story along the margin of Storglaciären. The combination of uniaxial compression (cone distribution about the compression axis) with simple shear (single maximum perpendicular to the shear plane for large strains) provides the clearest explanation for the split maximum (Kamb, 1972; Budd et al., 2013). Bouchez and Duval (1982), and Journaux et al. (2019) observe the tendency for the main c-axis maximum to spread, but not split entirely, in experiments using fixed plattens where compression could not be a factor. Li et al. (2000) attribute the spreading to transverse extension accompanying the flattening of the sample during deformation in their experiments. Two-dimensional numerical simulations by Llorens et al. (2016a, 2017) show this spreading and splitting occurs in simple shear with no flattening strain, and that it is enhanced by dynamic recrystallization. It is most pronounced at low strain rates. Qi et al. (2019) suggest that the spreading increases with increasing shear strain. In our case, at the margins of Storglaciären, the ice is deforming at high temperatures, low strain rates, and to high finite strains, consistent with conditions that enhance spreading in experiments (Qi et al., 2019) and in modeling (Llorens et al., 2016a, 2017). The degree of spreading and splitting is likely enhanced in these samples due to compression normal to the valley walls, in a direction normal to the shear plane, a pattern similar to that observed by Kamb (1972) and Budd et al. (2013).

The a-axes in both the low- and high-strain experiments of Qi et al. (2019) define a girdle parallel with the shear plane (Fig. 9). In the lower-strain experiments, the a-axes cluster mostly perpendicular to the shear direction (parallel to the vorticity axis), whereas in the higher-strain experiments they mostly cluster parallel with the shear direction (Fig. 9). This change in a-axis maximum from normal to the shear direction to parallel to the shear direction with increasing strain is also observed by Journaux et al. (2019), though there is not currently a good explanation for this switch. It is important to note that in both the experiments (Qi et al., 2019; Journaux et al., 2019) and in our study, the a-axis CPO indicates that slip is not isotropic in the basal plane (see Kamb, 1961).

In an attempt to mimic a possible grain sampling bias similar to that which we propose when dealing with warm coarse-grained ice, we randomly resampled subsets of 50 grains – allowing for random duplicates in the resampling (thus one grain may appear more than once in the resampling) – from the two warm experiments by Qi et al. (2019) at low and high strains and compared these to the stacked suite of natural samples in the same kinematic reference frame (Fig. 9). Subsets of the experimental data produce patterns that are more-diffuse and patchy than those for the full dataset and are broadly similar to patterns observed in natural coarse-grained ice. Importantly, the Qi et al. (2019) study does not suffer from grain sampling biases common to CPO characterization in warm glacial ice, due to the significantly finer and more consistent grain size (Fig. 9). Compared to the experimental results, the main c-axis maxima in the stacked data from our glacial ice samples (Fig. 8) are more elongate or "pulled apart" than those in the subsampled experimental data, and the girdle of a-axes is broader, with a cluster perpendicular to the shear direction, similar to the pattern observed in the lower strain experiments (Fig. 9). The more distinct c-axis sub-maximum in our combined data (Fig. 8) is offset from the main maximum in a synthetic sense with respect to the shear direction, rather than an antithetic sense as might be expected from the experimental data (Fig. 9). However, the less distinct sub-maximum, offset in the antithetic sense ~50º from the main maximum, is consistent with the secondary maximum in the experiments.

We interpret these results to mean that the grain sampling bias issue was not entirely resolved by making and combining composite sections, due to the very large grain size with interlocking shapes that still have not been entirely characterized. However, the overall similarity between the stacked data from composite sections from the three samples in the same kinematic reference (Fig. 8) to the CPO pattern presented by Qi et al. (2019) for fine-grained ice that has undergone low shear strains at high homologous temperature (Fig. 9, PIL91) suggest that the operative deformation mechanisms are similar.

It is important to note that we do not know the exact deformational history experienced by the ice in our natural samples, but the recent part of that history corresponds most closely to simple shear parallel to the ice margin. An additional similarity between the experiments (Qi et al., 2019) and the conditions of deformation experienced by our samples is that there is a small component of compression, which for our natural samples is perpendicular to the margins of the glacier, associated with the narrowing of the valley in the direction of flow (Fig. 10a). Thus our samples may represent similar kinematics to those in the experiments conducted by Duval (1981) and Budd et al. (2013) that involved simple shear combined with compression normal to the shear plane (Fig. 10b).

Hudleston (2015) calculated the finite shear strain required to rotate fractures towards parallelism with the flow direction along the margins of Storglaciären, and this indicated that the finite shear strain where we collected ice samples for our study is likely much greater than 2. This estimate exceeds the strain of the "high-strain" experiments done by Qi et al. (2019) and we might therefore expect our data to best match the "high-strain" experimental data. However, the a-axis pattern of our samples best matches the pattern for the "low-strain" experiments, suggesting a weaker effect of recrystallization on the CPO in nature than in the experiments. One possible reason for this comes from considering strain rate. In the experiments, shear strain rate was $\sim 10^{-4}\,s^{-1}$ whereas in natural ice along the south margin of Storglaciären, strain rate calculated from velocity measurements (Hooke et al., 1983b; Hooke et al., 1989) and modeling (Hanson, 1995) is $\sim 10^{-10}\,s^{-1}$. Dynamic recrystallization and grain growth are effective at low strain rates (Hirth and Tullis, 1992; Takahashi, 1998; Qi et al., 2017). They may also be enhanced under high temperature, low stress conditions, as shown by Cross and Skemer (2019) using empirical data, although these authors note that this conclusion needs testing because it is counterintuitive. In any case, both grain boundary mobility (function of temperature) and driving force (function of the storage of dislocations as a result of stress) are important, and the scaling between these two from experiment to natural conditions is not known. With the high finite strain experienced by our samples the ice must be completely recrystallized, with further strain producing further recrystallization. Considering all of this, it may be the case that recrystallization in nature is intense at the high finite strains encountered, and modifies the CPO so that it does not attain the degree of development found in the experiments. The resulting CPO (Fig. 10b) will then likely reflect only the latest part of the deformational history, being continually modified by dynamic recrystallization as deformation continued.

### 7 Conclusions

By developing a new sample preparation method to create composite sections for each sample collected, we are able for the first time to use cryo-EBSD to obtain complete (c- and a-axes) crystallographic orientation measurements for interpreting CPO patterns in natural, coarse-grained glacial ice subjected to simple shear, for

the marginal ice of Storglaciären. A single composite section captures a relatively large number (~50-100) of grains, in our case from an ice sample of ~200mm x 150mm x 75mm dimensions and with >20mm grain size, and combining composite sections from adjacent samples increases further the number of grains sampled. The larger number of grains in this new approach allows us to better characterize CPO patterns in coarse-grained ice than has been done previously, and it sheds new light on the significance of microstructural processes associated with previously identified multimaxima CPO patterns. Specifically, we conclude that a grain sampling bias of interlocking, large (>20mm) branched crystals that appear multiple times as apparent island grains in thin section contributes to the apparent multiple maxima CPOs displayed in our natural ice samples. We have not removed this effect, but confirmed it using both c- and a- axes, and partly compensated for it by increasing the effective sample size. Such bias also certainly contributed to similar CPOs that have long been identified in other studies of natural, warm, coarse-grained ice. Without better establishing 3D grain size and shape, it will be difficult to fully eliminate or account for this bias, but a combination of systematic sampling, composite sample preparation, and data stacking will help more accurately define CPOs.

We predict that from our study and from a comparison with experimental results, a fully representative CPO, if enough data from a large enough volume of ice were sampled, would consist of: 1) a c-axis CPO with one maximum that is extended or "pulled apart" in a plane perpendicular to the shear direction, and a weaker maximum 45º-60º from the shear plane; and 2) a broad girdle of a-axes parallel to the shear plane with a cluster perpendicular to the shear direction, reflecting non-isotropic slip within the basal plane. Such a pattern assumes that the dynamic recrystallization of ice deformed to high finite strains, under slow strain rate and high temperature conditions results in the observed large grain size and resetting of CPO to reflect the local kinematic conditions.

Our new sample preparation method allows for faster, and more accurate collection of complete crystallographic orientation data and microstructural analyses of coarse-grained ice. This opens a range of opportunities for further analyses to aid in the understanding of micromechanical processes governing rheological properties of such ice. Future work will benefit from better quantification of 3D grain size and shape to help improve the sample preparation methods in order to minimize any grain sampling bias. Additionally, more work should be done to quantify the effects of dynamic recrystallization in the context of shear strain along the margins of glaciers and should be taken into account when assessing these CPO patterns.

**Appendix A**

CPO representations using modern techniques, such as AITA or EBSD, are often plotted as all-pixel orientations. Gagliardini et al. (2004) demonstrated that the weighted area procedure based on all-pixel orientations is a statistically better representation of the CPO if the volume of ice measured is fully representative, such that results from two samples of that volume are the same. This assumes that the measured area of a grain in cross section is the mean projected area of the grain. All-pixel orientations are especially important for providing estimates of bulk physical properties, such as the polycrystal elasticity tensor, and may take into account the issue of parent grains with satellite island grains, although this will only be true if the sample is large enough to contain a sufficient number of grains to provide a truly representative CPO. In addition, the presence of island grains calls into question the assumption that the measured area of a grain in cross section is the mean projected area of the grain. If there are no repeat grains in a representative sample,

one-point-per-grain plots would yield the same CPO patterns (e.g. small circle girdle in fig. A1), although the eigenvalues of the orientation tensor, if calculated, will generally differ. If the volume of ice measured is not fully representative, one-point-per-pixel will bias CPO patterns towards larger grains and is a sure way to produce a multimaxima CPO. This produces unrepresentative grain area biases, as these large grains may not be

the largest or close to largest grain in a representative volume of ice. The one-point-per-grain method will reduce that area bias, but could have repeat grains and bias results towards finer grains if the fine grains differ in CPO from the large grains (unlikely in the case of Storglaciären). One-point-per-grain analyses, therefore, represent the CPO pattern better when you have less representative samples (fig. A1). This is true in our case, in which we have a broad range of grain sizes in 2D and few grains. In addition to better representing the CPO

pattern, plotting the one-point-per-grain method allows for direct comparison with the fabrics described in earlier studies when all the data were represented this way.

The issue of statistics is not straightforward for coarse-grained ice with the existence of multimaxima CPOs. Any way of attempting to eliminate the effects of multiple counting of individual grains that appear more than once in a thin section or in multiple sections intersecting a single crystal would be ad hoc. Doing statistical

tests while ignoring this phenomenon is of little use. Kamb's (1959) method of contouring provides a way of establishing the statistical significance of maxima in a fabric, but this is only meaningful if multiple points from the same grain are excluded. We believe that use of eigenvalue methods and associated statistics is inappropriate for multiple maxima fabrics.

**Acknowledgments.** We are grateful to the University of Stockholm and the Tarfala Research Station for making this field work possible and providing us with the tools necessary to access the glacier and collect samples, to Troy Zimmerman for his field assistance and to Hannah Blatchford for her help transporting samples to New Zealand and aiding in sample preparation. Thoughtful and helpful reviews were provided by M. Montagnat and A. Tomassi. This research was made possible by funding provided by Graduate Student

Research Grant administered by the Geological Society of America, and Grant-in-Aid of Research administered by Sigma Xi, the Scientific Research Society. Microscopy in Otago was funded through Marsden Grant UOO052 to Prior.

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

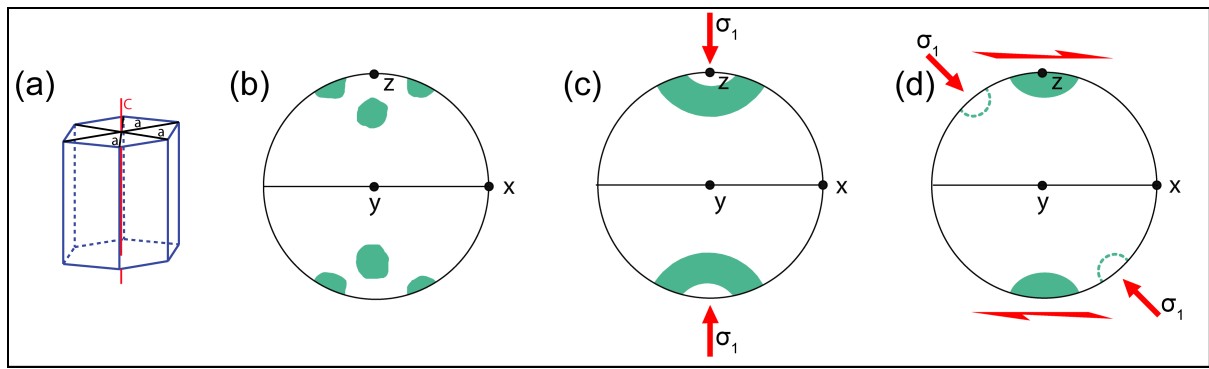

**Figure 1. Schematic image showing (a) an ice crystal and its defining c- and a-axes; (b) multimaxima fabric pattern identified in warm, coarse-grained glacial ice. The x, y, z axes define the symmetry, but the kinematics are debated (c)**

1025   **a small circle girdle or cone shape in uniaxial compression and (d) a strong single maximum fabric ± a weaker second maximum in simple shear. (c) and (d) are from experiments and (c) (arguably) and (d) also from nature, and the axes x, y, z define the kinematic reference frame. CPO plots here and in subsequent figures are all equal-area, lower hemisphere projections. In (b) and (d), z is normal to the foliation, and x is parallel to the shear direction. There is no shear plane implied in (c).**

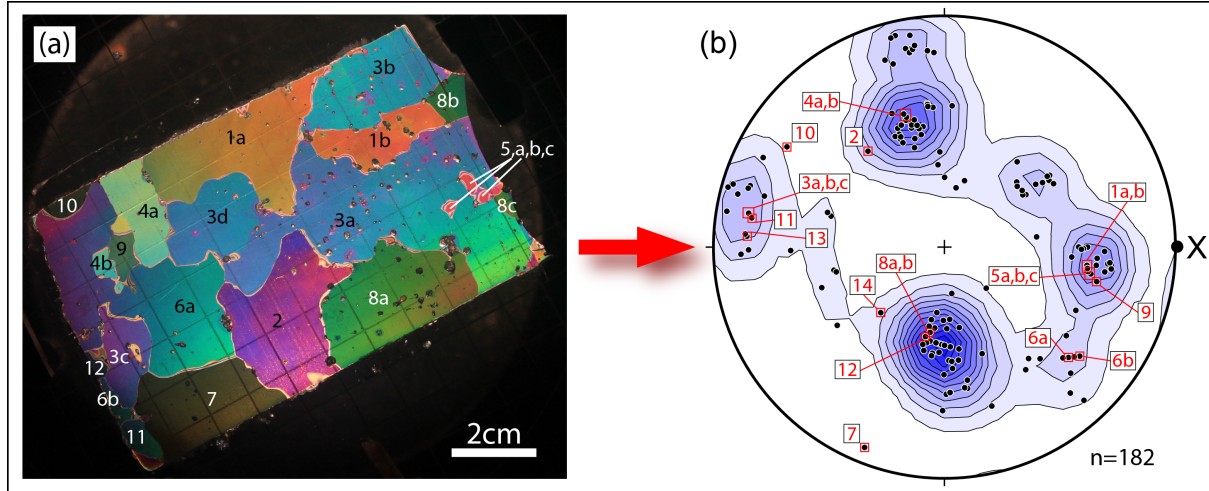

**Figure 2. (a) A thin section under cross-polarized light from sample SG6-B collected in 2016. Grains are labeled based on their c-axis orientations, measured using a universal Rigsby stage. Grains with the same orientation were tentatively marked as the same grain as indicated by lettering. Color gradients across some larger grains are a result of inconsistent thin section thickness; (b) Associated c-axis plot compiled from 8 thin sections from the sample SG6-B. This plot contains orientations of all grains measured, and is contoured using the Kamb method (Kamb, 1959). When possible duplicates within the same section are removed, the pattern maintains its multimaxima nature, but is weaker (not shown). Numbered data points correspond to numbered grains in (a). In order to obtain 100 individual crystal measurements, 5-15 thin sections had to be made for each sample, depending on the overall grain size in the individual sample. The projection is plotted such that the pole to foliation is vertical and x is the flow direction. The data in this figure are not included in the combined figure 8 because after U-stage work, there was not enough of this sample to use for EBSD, and thus there are no a-axis data available.**

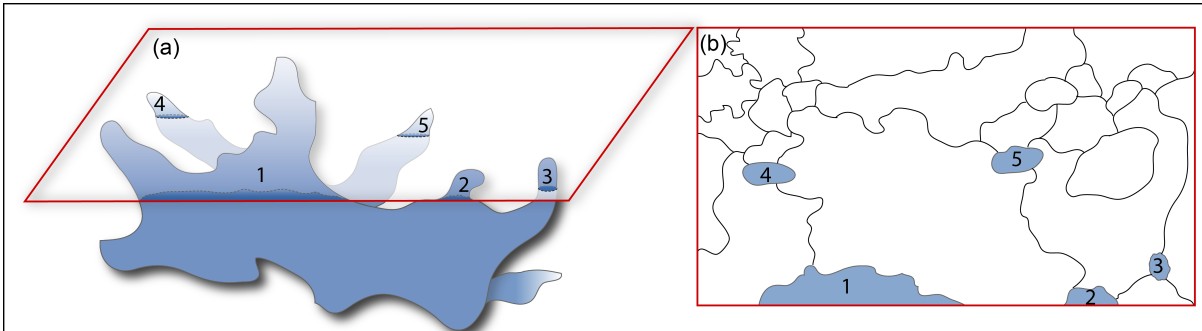

**Figure 3. (a) Schematic branching crystal and associated (b) two-dimensional thin section highlighting the notion of island grains that would appear in a two-dimensional thin section. Numbered branches in three dimensions correspond to the numbered island grains in two-dimensions. Part of the grain in (a) lies above the section and part below.**

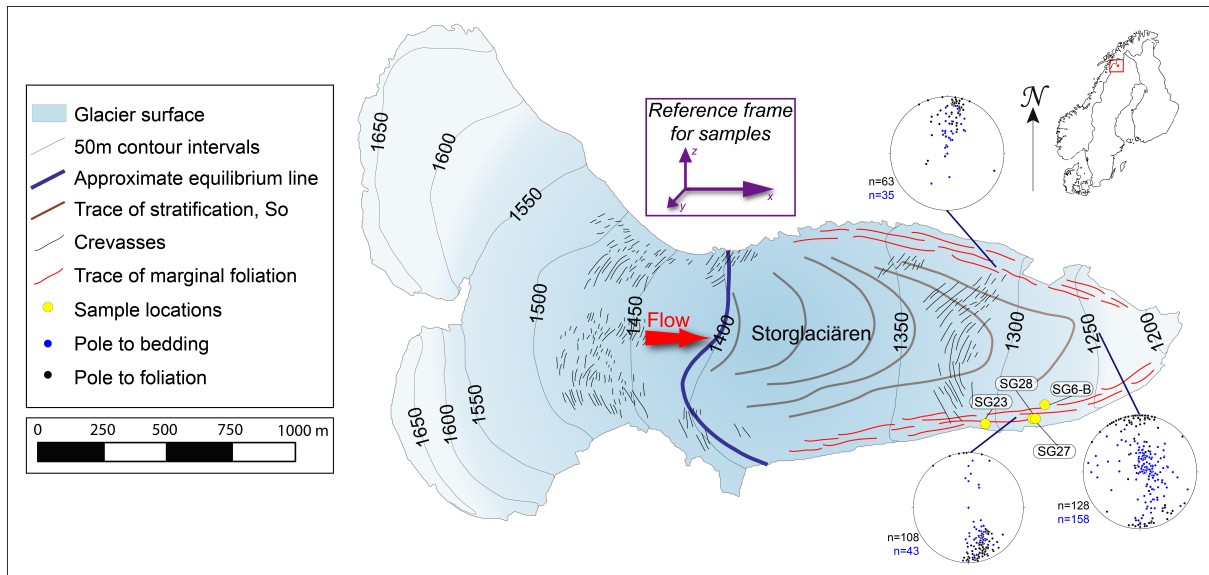

Figure 4. Simplified map of Storglaciären highlighting the traces of structural elements and orientations of foliation and bedding for the north margin, south margin, and center of the glacier in the ablation zone. Locations of samples SG6-B, SG23, SG27 and SG28 are labeled. The orientation diagrams of planar fabric elements (stratification and foliation) are in geographic coordinates. The sample reference frame for the remainder of the paper is represented, where x is the flow/shear direction y is the vorticity axis and z is north.


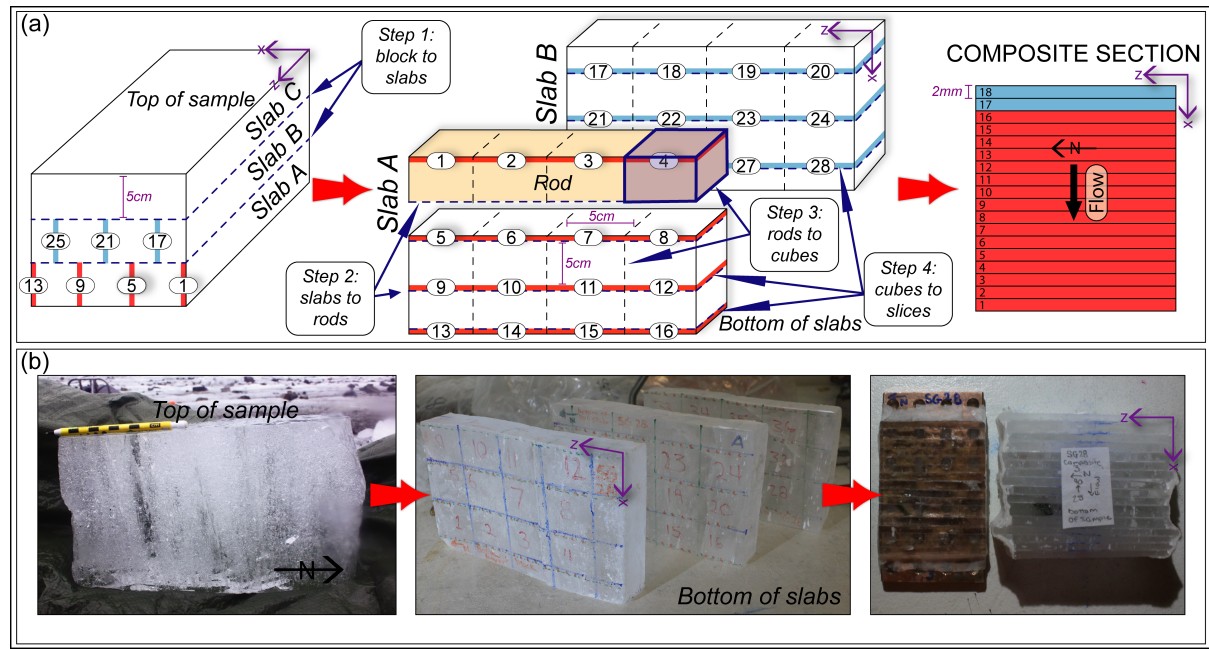


Figure 5. (a) Schematic sample preparation for one composite section. Steps for each cut to progress from sample to slice are specified with dashed lines, and examples of each are highlighted with dark blue arrows. Individual 2mm slices are shown throughout the process on slab A and slab B, and the numbers correspond to the unique slice in the final composite stack. At least 36 slices were cut from every sample (using slabs A, B and C) to construct composite sections. At least two composites for each sample comprised the final data set (1 composite comprised of slices 1-18 is shown in figure). (b) Sample preparation illustrated using SG28 from block sample to composite section. Note that the number sequences varied from sample to sample so the numbers on the slabs do not match the schematic in (a). Composites are oriented in a kinematic reference frame such that x is the flow/shear direction, y is the vorticity axis and z is north.



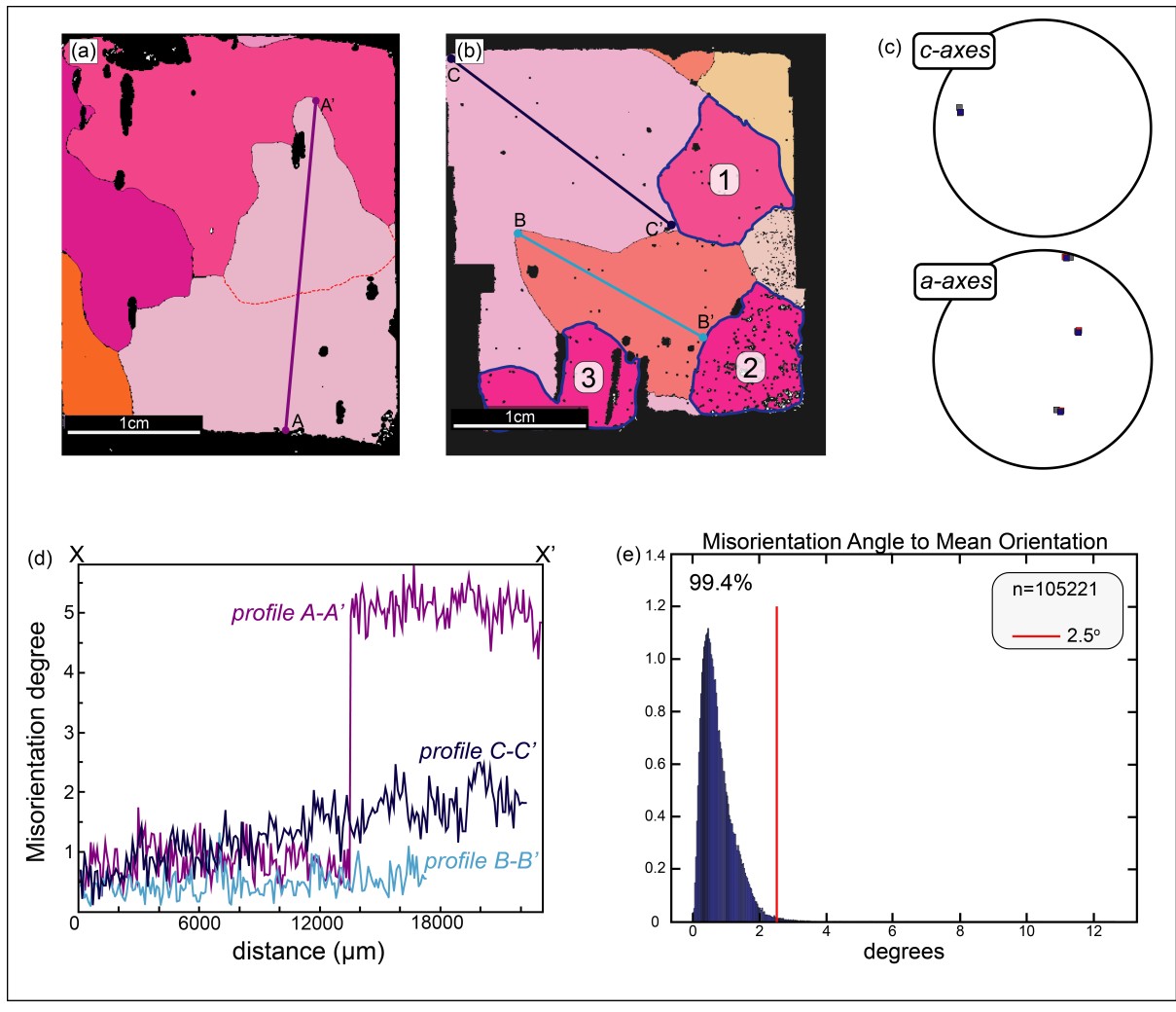

**Figure 6. A typical whole section EBSD analysis with: (a) and (b) maps of whole sections from SG28. (a) Subgrain boundary is shown with a red dashed line, and profile A-A' crosses that boundary, (b) highlighting three potential island grains within the section, and profile lines B-B' and C-C' across two grains; (c) one-point-per-grain c- and a-axes of the highlighted grains 1, 2 and 3 in B. (d) misorientations profiles relative to first pixel along A-A', B-B' and C-C'; (e) Misorientation angle of each pixel in EBSD map (b) with respect to the mean orientation of the grain. The red line highlights that 99.4% of the misorientations between pixels lie below 2.5º.**


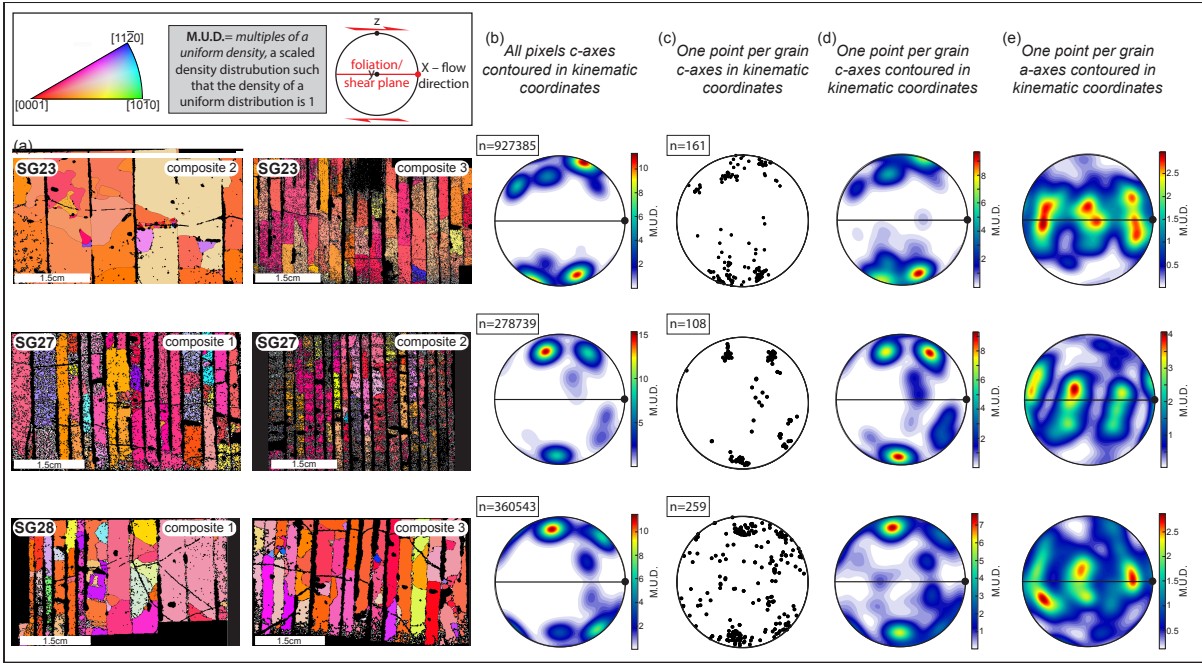

**Figure 7. EBSD maps and associated CPOs for composite sections from samples SG23, SG27 and SG28. Data from each pair of composites are combined to give the bulk CPO for each sample. (a) EBSD images of the two composite sections, where the vertical black lines represent junctions between individual slices of the composite. (b) contoured plots of c-axis orientations of all pixels; (c) uncontoured plots of c-axis orientations representing one-point-per-grain; (d) contoured plots of c-axis orientations representing one-point-per-grain; and (e) contoured a-axis orientations representing one-point-per-grain. All plots are in a kinematic reference frame where x is the shear flow/shear direction (black dot), y is the vorticity axis and z is north.**

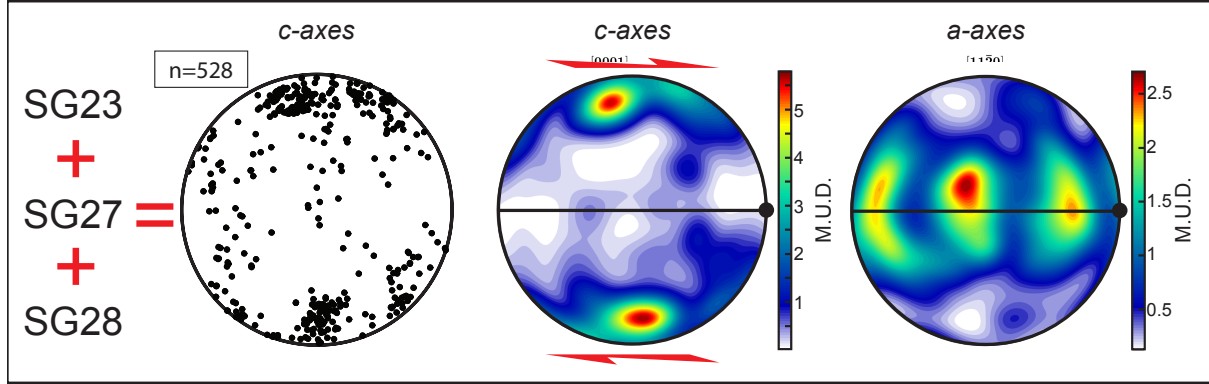

**Figure 8. Combined data (one-point-per-grain) from SG23, SG27 and SG28 in the same kinematic reference frame. Plots from left to right show c-axes uncontoured, c-axes contoured and a-axes contoured.**

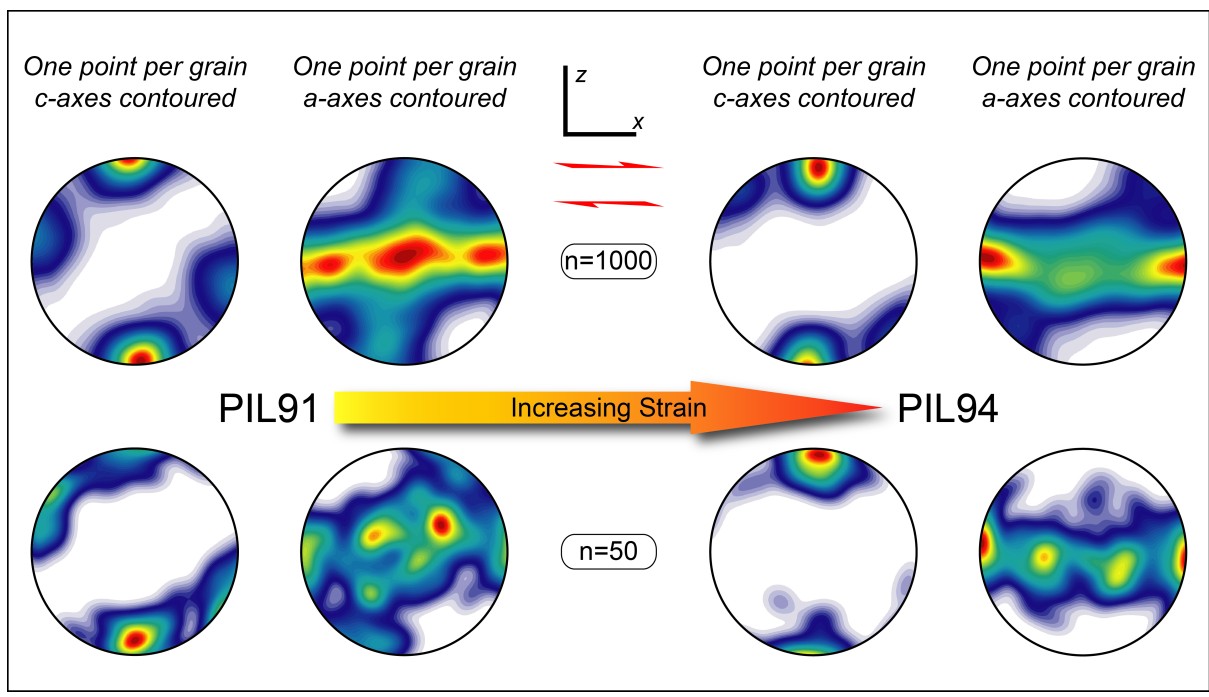

**Figure 9.** Samples PIL91 (γ= 0.62) and PIL94 (γ= 1.5) from Qi et al. (2019), displaying the full data set, and a typical subset of randomly resampled data. Both of these samples were deformed at -5ºC at 20MPa confining pressure in a cryogenic gas medium apparatus under constant axial displacement rates and terminated at different shear strains. In all cases, the pole figures are in kinematic coordinates, with x being the shear direction and x-y the shear plane.


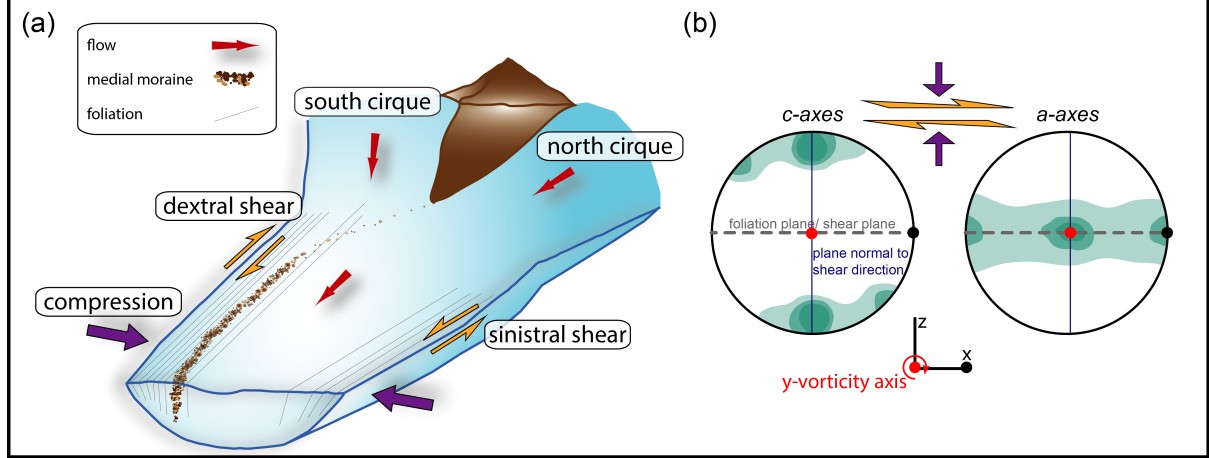


**Figure 10. (a)** Schematic of the combination of simple shear and compression experienced by the ice as the valley narrows (similar to Storglaciären) and **(b)** associated schematic CPO plots (for marginal ice), highlighting the relationship of the CPO to the foliation/shear plane (gray), the vorticity axis (red) and the plane normal to the shear direction (navy blue), expected for dynamic recrystallization at low strain rates and high temperatures.

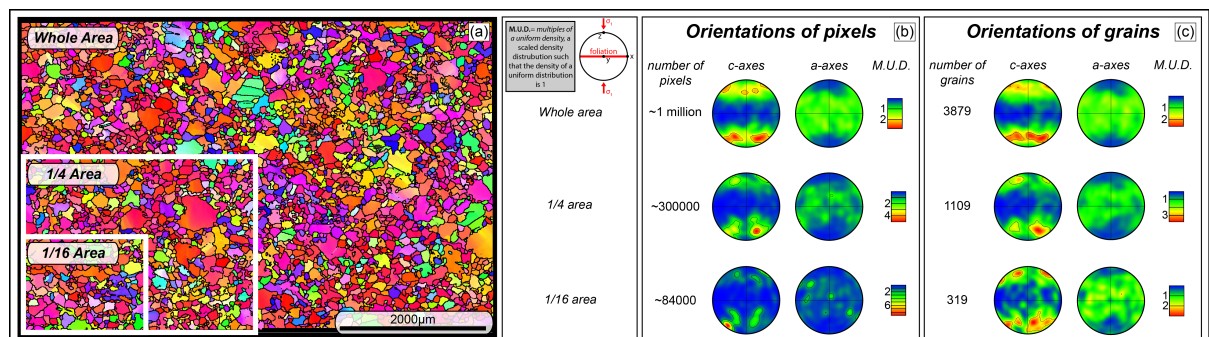


**1110**      **Figure A1: EBSD map and associated CPO plots for sample PIL36 deformed in uniaxial compression at -9.8°C from Qi et al. (2017), highlighting the difference in representing data as all-pixel orientations vs. one-point-per-grain orientations as a function of sample size. (a) EBSD map, with boxes representing subsampled areas. (c) CPO plots (c- and a-axes) of all pixel orientations from the entire sample area, ¼ the sample area and 1/16 the sample area. (d) CPO plots (c- and a-axes) of one-point-per-grain orientations from the entire sample area, ¼ the sample area and 1/16 the sample area.**

**1115**