# Peer review of "Full crystallographic orientation (c- and a-axes) of warm, coarse-grained ice in a shear dominated setting: a case study, Storglaciären, Sweden"

_The Cryosphere, 2020_

## Referee Comment (RC1) · Maurine Montagnat (Referee) · 2 Jul 2020

Review of paper by Monz et al. 2020 Title: Electron backscatter diffraction (EBSD) based determination of crystallographic preferred orientation (CPO) in warm, coarse-grained ice: a case study, Storglaciären, Sweden Author(s): Morgan E. Monz et al. MS No.: tc-2020-135 MS Type: Research article The Cryosphere

The paper, entitled "Electron backscatter diffraction (EBSD) based determination of crystallographic preferred orientation (CPO) in warm, coarse-grained ice: a case study,

[Figure]

Storglaciären, Sweden" suggest a method adapted to estimate the crystallographic preferred orientation (CPO) of samples with very large grains extracted from the margin of a glacier located in Sweden. The adapted method aims to respond to the well known limitation for good statistical CPO measurements owing to a too limited number of grains on classical thin sections or EBSD samples. The main conclusions of the paper are first that multimaxima CPO classically observed in large grain samples are very likely due to a too limited number of grains in the sample area studied, and then that grain boundary migration may dominate the recrystallization processes in the conditions encountered in the glacier.

The method presented in this paper, by creating a composite sample out of a large block of ice, seems to be suitable to increase the number of different orientations measured in one single EBSD sample of limited size. The scientific conclusions provided in the paper are not new and far too weak for the expectations of a standard research paper in The Cryosphere. Therefore I would suggest to reject the paper in TC, but I would encourage the authors to submit it in a journal with a specific section for "Instrument and methods", deepening the method description, and increasing the work to validate it. In the following, I give more detailed explanations for this evaluation.

- The method described, although sounding interesting, is not compared to any other type of measurements (for instance, many sample analyses over a continuous part of a core, or of a block of ice in order to provide enough grains for a good statistics) in order to assess its statistical representativity and it robustness. For instance, are we sure not to measure several times the same large crystal coming from the depth of the block, since some crystals are more than 90mm large? Owing to the fact that exact shape and location of grains are lost, there is no way to verify such a situation, as is done in figure 2 for instance. The introduction pretends that the use of a-axes measurements could provide supplementary information to check the belonging of measured areas to one single crystal or several, but this procedure is not used neither described later... There is only a very weak discussion about the orientation error produced by this multi-slicing

technique... Although it could be quite strong, and add on at each slicing step.

- The too limited number of measured crystals is attributed solely to the measurement techniques (AITA or EBSD) imposing too small samples. This is not so true since it is possible to measure several contiguous samples from a ice core (see Dahl-Jensen et al. 2013, Montagnat et al. 2014 for instance were analyses along contiguous samples from 1 m long core sections were done). In many cases, the limitation is due to the limited size of the core extracted.

- One of the main conclusions is related to the observation, in some previous studies, of multi-maxima, and there attribution to a too limited number of crystals. This result is not so new and was intuited by most of the authors responsible for the mentioned studies. Experimental observations (such as the ones from Qi et al. 2019 shown in the paper) enabled to confirm this intuition already since, with a larger number of grains, the multi-maxima texture do not exist anymore. Although there is one dominating orientation in the combination of samples presented on figure 8, the multi-maxima remains, with 3 main orientations. So the result is not so obvious and can not lead to such a firm conclusion. Moreover, to be more affirmative, one would have needed more results, on various samples, what is not shown in this study.

- The other conclusion related to the grain boundary migration dominating dynamic re-crystallization processes in the studied conditions is not new at all, and simply confirm the observations by most authors working on dynamic recrystallization mechanisms in warm conditions (see for instance De la Chapelle et al. 1998, but also the laboratory work by Jacka and co-authors, or the most recent work by Journaux et al. 2019).

- I would like to add another important comment related to the references provided in the text. Although the authors used the review papers by Faria and co-authors (2014), it is necessary to provide the references of the original works to whom the credit should be given. Otherwise, the community will little by little loose track of these original works, and the credit will go only to the one who wrote the review. This is not fair, and also

not respecting the ethical rules of citations in publications. For instance, line 50, p2: you could cite Van der Ween and Whillans, 1990, Mangeney et al. 1997 among others (original citations are provided in Faria et al. 2014). line 62, p2, Russell-Head et al. 1979, Azuma and Higashi 1985, Alley 1988, Gow et al. 1997, Castelnau et al. 1998, etc. would be more appropriate. Again line 117, p3

- About open cones CPO in polar bore holes. Once again, the citation of Faria et al. 2014 is inappropriate, since Faria and co-authors did not make any measurement along deep ice cores, and this is not true, I think, that this type of CPO is not observed along polar ice cores. Also less clear than in experimental work, some CPO very close to open cones are observed in the bottom of the GRIP, GISP2, BYRD cores for instance.

- Line 118: the multi-maxima CPO is not enigmatic and some hypothesis were given by different authors... See for instance De La Chapelle et al. 1998.

- Part 6.3: lines 430 to 436, the dynamic recrystallization processes are mentioned as a likely difference between the experimental and natural conditions, owing to the difference in strain rate. Although already in the experimental conditions is dynamic recrystallization very active (especially at this high temperature), and the driving force for GBM is even stronger since it is associated to the storage of dislocations at GB, the latter being expected to be stronger at high relative strain rate. At lower strain rate, we expect the dislocation storage to be slower relative to GB mobility.

N. Azuma and A. Higashi. Formation processes of ice fabric pattern in ice sheets. Ann. Glaciol., 6:130– 134, 1985. O. Castelnau, H. Shoji, A. Mangeney, H. Milsch, P. Duval, A. Miyamoto, K. Kawada, and O. Watanabe. Anisotropic behavior of GRIP ices and flow in Central Greenland. Earth and Planetary Science Letters, 154(1-4):307 – 322, 1998. S. de la Chapelle, O. Castelnau, V. Lipenkov, and P. Duval. Dynamic recrystallization and texture development in ice as revealed by the study of deep ice cores in Antarctica and Greenland. J. Geophys. Res., 103(B3):5091–5105, 1998. Dahl-Jensen, and co-authors. Eemian interglacial reconstructed from a Greenland folded ice core. Nature,
493:489–494, 2013. M. Montagnat, N. Azuma, D. Dahl-Jensen, J. Eichler, S. Fujita, F. Gillet-Chaulet, S. Kipfstuhl, D. Samyn, A. Svensson, and I. Weikusat. Fabric measurement along the NEEM ice core, greenland, and comparison with GRIP and NGRIP ice cores. The Cryosphere, 8(4):1129–1138, 2014. D. S. Russell-Head and W. F. Budd. Ice-sheet flow properties derived from bore-hole shear measurements combined with ice-core studies. Geol. Soc. Australia, 64:159, 1979.

---

## Referee Comment (RC2) · Andrea Tommasi (Referee) · 10 Jul 2020

The article presents EBSD data on the crystallographic preferred orientations of ice in a shear zone composing the lateral boundary of a warm mountain glacier, where deformation took place at temperatures > -10°C. The main conclusion of the article is that under natural low strain rate conditions simple shear under high homologous temperature conditions, which favor dynamic recrystallization by grain boundary migration, produces ice CPO evolutions similar to those observed in simple shear experiments at high homologous temperature conditions. A secondary conclusion is that the multimaxima CPO patterns observed in previous samplings of natural high-temperature shear zones in ice were an artifact due to analysis of a too small number of grains and oversampling of some grains with very irregular shapes. Both conclusions seem undoubtedly correct, but not really new.

The main challenge in collecting such data was the extremely coarse grain sizes of ice in such conditions (typically > 20mm, but attaining >90mm, as the authors describe). To circumvent this problem, the authors analyzed composite sections constructed from serial sectioning of blocks 15x15x30 cm blocks. The technique is presented as new, but use of multiple coherently-oriented samples to analyze a representative volume of coarse-grained materials is a rather traditional (and effective) solution for this problem. Given the fact that the authors still observe multiple maxima CPO patterns for all analyzed composites, one may question if the technique proposed is really effective (the spacing used for the sectioning is probably still smaller than the maximum dimension of the grains). One may therefore question why should one prefer this method to the even more traditional one (at least in geology) of collecting oriented samples in a series of profiles normal to the shear zone trend and then add up the data for samples with similar positions across the shear zone. This second approach would allow to: (1) spread the sampling a much larger volume, (2) preserve the relation between CPO and microstructure, which is essential for discussing the role of deformation and recrystallization processes on the evolution of the CPO, and (3) collect data for variable finite strains (which is missing here and would have been extremely useful to discuss some features, such as the deviation of the [0001] maxima relatively to the normal to the shear plane along the plane normal to the shear direction or how the CPO evolves with finite strain).

In conclusion, neither the results nor the technique are completely new. If the article is to be published (I do not know the journal well enough to make a recommendation), it has to be revised to present in a more objective way its actual contribution: new data on the evolution of CPO of ice in natural shear zones, which confirm the current knowledge

on the subject: simple shear under high homologous temperature produces a CPO characterized by concentration of [0001] axes normal to the shear plane. Moreover, the discussion should be reinforced and present a comparison of the observations with all available experimental data in simple shear (why focus the comparison on a single set of experiments?) discussing in a more straightforward way the similarities and discrepancies between the different datasets. The rather 'surprising' observations of: (1) lack of a maximum of <a>-axes parallel to the flow direction and (2) the deviation of the [0001] maxima relatively to the normal to the shear plane along the plane normal to the shear direction - should be discussed in a more effective way. The present discussion, although long, does not propose any clear explanation for neither of the two observations.

Additional points: The statements presenting the relation between microphysical processes and CPO evolution in the abstract, introduction, discussion, and conclusion lack precision and give the (false, in my point of view) impression that CPO evolution is mainly controlled by recrystallization (cf. lines 15 & 58-60) or that dynamic recrystallization may completely reset the CPO (cf. lines 30-32 & 434-436). As I see CPO is produced by dislocation glide and recrystallization modifies it, by creating new orientations (most often only dispersion around the orientation of the parent grains) and selectively consuming others when grain growth is effective as it is the case here. The first process certainly buffers the increase in the CPO intensity, but not fully resets the CPO. The second may significantly change the CPO when grain growth is orientation-dependent.

Which are the arguments which justify that low strain rates should enhance dynamic recrystallization and grain growth (l. 434)? I would rather propose the opposite as the forces associated with dislocation density gradients should be smaller at low strain rates.

Referencing is often loose and there are many places where pertinent references are missing. For instance, l. 61, Wenk and Christie (1991) is not the best reference in

a phrase dealing with CPO-induced mechanical anisotropy when there are a large number of studies that investigated precisely this effect (cf. review by Gagliardini et al. 2009 and references therein).

The aims of the article should also be redefined. Those stated in l.79-82 were probably the initial aims of the study, but given the results, they cannot be the aims of the article.

The authors indicate that 8 areas were sampled and that at least two composite sections were made for each of the eight samples. However, in the map only 4 sampling sites are located and data is shown for only 3 samples. Why? Where are the data for SG6-B, which seems from its location in the map to sample a lower strain domain? Data for domains with variable finite strains may help explaining the two unusual features in the observations: (1) the lack of a maximum of <a>-axes parallel to the flow direction and (2) the deviation of the [0001] maxima relatively to the normal to the shear plane along the plane normal to the shear direction.

In l. 294, it is indicated that EBSD work is performed on 40mm x 60mm sections. However, all EBSD maps presented in the article are much smaller (25 x 25 mm on average in Fig. 6a and 3.5 x <3 mm in Fig. 7a). Why use a reduced analysis area in a study where the size of the mapped area is critical?

---

## Author Comment (AC2) · 28 Aug 2020

Response to Referees' Comments

We are grateful for the referees in pointing out where more information should be provided and where clarification is needed. In the responses below, we include additional information to bolster claims made in the manuscript. We disagree with the referees on a number of points, as indicated.

These responses are provided in the order in which the referees' comments were made. These responses are provided in the order in which the referees' comments were made. The referee comments are italicized.

Referee No. 2

1) *To circumvent this problem* (of very large grain size), *the authors analyzed composite sections constructed from serial sectioning of blocks 15x15x30 cm blocks. The technique is presented as new, but use of multiple coherently-oriented samples to analyze a representative volume of coarse-grained materials is a rather traditional (and effective) solution for this problem.*

We do not claim that using multiple coherently-oriented sections to obtain a sufficiently large sample size for coarse-grained ice is new, our method of doing this is new, which makes it practical for EBSD work.  See response #2 to the first referee.

2*) Given the fact that the authors still observe multiple maxima CPO patterns for all analyzed composites, one may question if the technique proposed is really effective (the spacing used for the sectioning is probably still smaller than the maximum dimension of the grains). One may therefore question why should one prefer this method to the even more traditional one (at least in geology) of collecting oriented samples in a series of profiles normal to the shear zone trend and then add up the data for samples with similar positions across the shear zone. This second approach would allow to: (1) spread the sampling a much larger volume, (2) preserve the relation between CPO and microstructure, which is essential for discussing the role of deformation and recrystallization processes on the evolution of the CPO, and (3) collect data for variable finite strains (which is missing here and would have been extremely useful to discuss some features, such as the deviation of the [0001] maxima relatively to the normal to the shear plane along the plane normal to the shear direction or how the CPO evolves with finite strain).*

Doing what the referee proposes would be impractical for the situation encountered in this and most valley glaciers. We do not have clear markers of shear strain that allow us to document a strain gradient across the marginal ice, but we can reasonably assume that fairly closely spaced samples come from a homogeneously deformed volume of ice. Collecting, handling, transporting and preparing for analysis many more large samples was beyond the resources available to us. We could not do on a large scale what Hudleston (1977) was able to do on the scale of a single thin section (and what the referee suggests here) for a small-scale shear zone in cold ice at the margin of the Barnes Ice Cap. However, more systematic sampling, allowing for individual slices of each composite to be spaced by >15cm would be beneficial and appropriate for a follow-up study.

3) *In conclusion, neither the results nor the technique are completely new. If the article is to be published (I do not know the journal well enough to make a recommendation), it has to be revised to present in a more objective way its actual contribution: new data on the evolution of CPO of ice in natural shear zones, which confirm the current knowledge on the subject: simple shear under high homologous temperature produces a CPO characterized by concentration of [0001] axes normal to the shear plane.*

We refer to the response #1 of the first referee to emphasize what is new in our study. We should note that all natural ice deformation is under conditions of high homologous temperatures. There is almost no new data for the evolution of CPO of natural ice in shear zones, because there is very little close control of strain gradients in natural ice. Nearly all the published data comes from laboratory experiments. As far as we are aware there is still only one study of fabrics in natural ice constrained to be from a well-defined shear zone (Hudleston, 1977).

4) *Moreover, the discussion should be reinforced and present a comparison of the observations with all available experimental data in simple shear (why focus the comparison on a single set of experiments?)*

As far as we are aware there are only two published sets of experiments that document both c-axis and a-axis fabrics in simple shear in ice, and those are the ones by Qi et al. (2019) and Journaux et al. (2019), both of which we cite and the results of which we compare with our data. In the case of Qi et al. (2019), we show how taking a subset of the data leads to less well defined fabric patterns that might be compared to natural fabric patterns with limited grain counts. We do cite other experiments done in simple shear or in simple shear plus compression normal to the shear plane, but the data in these is not presented in way that allows for direct comparison with our data or the data of Qi et al. (2019).

5) *The rather 'surprising' observations of: (1) lack of a maximum of <a>-axes parallel to the flow direction and (2) the deviation of the [0001] maxima relatively to the normal to the shear plane along the plane normal to the shear direction - should be discussed in a more effective way. The present discussion, although long, does not propose any clear explanation for neither of the two observations.*

We do not have a good explanation for the first point here, and the switch with increasing strain from a-axes perpendicular to flow at low strain to parallel to flow at high strain was not explained by Qi et al. (2019) or Journaux et al. (2019) in their experiments. The second point we do discuss (l. 379-385) though perhaps could do so more effectively. The deviation or spreading of the main [0001] maximum in a plane normal to the shear plane and in a direction perpendicular to the shear direction (see response #1, fig. R2) is found both in simple shear experiments (Kamb, 1972; Bouchez and Duval, 1982; Journaux et al., 2019) and in experiments involving simple shear with the added effect of compression or flattening normal to the flow plane (Kamb, 1972; Budd et al., 2013; Li et al., 2000). The combination of uniaxial compression (cone distribution about the compression axis) with simple shear (single maximum perpendicular to the shear plane for large strains) provides the clearest explanation for the split maximum (Kamb, 1972; Budd et al., 2013). Bouchez and Duval (1982), and Journaux et al. (2019), however, observe the tendency for the main c-axis maximum to spread in experiments using

fixed plattens where compression could not be a factor. Li et al. (2000) attribute the spreading to transverse extension accompanying the flattening of the sample during deformation in their experiments. Numerical simulations by Llorens et al. (2016a, 2017) show this spreading does occur in simple shear with no flattening strain, and that it is enhanced by dynamic recrystallization. It is most pronounced at low strain rates. Qi et al. (2019) suggest that the spreading increases with increasing shear strain. In our case, at the margins of Storglaciären, the ice is deforming at high temperatures, low strain rates, and to high strain, consistent with conditions that enhance spreading in experiments and in modeling.

6) *The statements presenting the relation between microphysical processes and CPO evolution in the abstract, introduction, discussion, and conclusion lack precision and give the (false, in my point of view) impression that CPO evolution is mainly controlled by recrystallization (cf. lines 15 & 58-60) or that dynamic recrystallization may completely reset the CPO (cf. lines 30-32 & 434-436). As I see CPO is produced by dislocation glide and recrystallization modifies it, by creating new orientations (most often only dispersion around the orientation of the parent grains) and selectively consuming others when grain growth is effective as it is the case here. The first process certainly buffers the increase in the CPO intensity, but not fully resets the CPO. The second may significantly change the CPO when grain growth is orientation dependent*

We agree entirely with how the referee interprets the CPO and thought that is what we stated in the manuscript. We apparently have given a false impression. We will attempt to clarify.

7) *Which are the arguments which justify that low strain rates should enhance dynamic recrystallization and grain growth (l. 434)? I would rather propose the opposite as the forces associated with dislocation density gradients should be smaller at low strain rates.*

See the response to point #14 of the first referee.

8) *Referencing is often loose and there are many places where pertinent references are missing. For instance, l. 61, Wenk and Christie (1991) is not the best reference in a phrase dealing with CPO-induced mechanical anisotropy when there are a large number of studies that investigated precisely this effect (cf. review by Gagliardini et al. 2009 and references therein).*

This is a fair point, which can be addressed, although we believe that Wenk and Christie (1991) is an important reference as these authors discuss the effect of CPO on the internal flow strength of rocks (relating back to the many important purposes for studying ice l. 42-45). Examples of additional references, relating specifically to CPO development modifying the internal flow strength of polycrystalline ice include: Steinemann, 1958; Lile, 1978; Pimienta and Duval, 1987; Alley, 1988; Alley, 1992; Azuma and Azuma, 1996; Gagliardini, 2009.

9) *The aims of the article should also be redefined. Those stated in l.79-82 were probably the initial aims of the study, but given the results, they cannot be the aims of the article.*

We are puzzled by this comment. The aims of the study are as given in lines 79-82. The only thing we might change is to replace the word "fully" by "better," since we have not fully addressed the issue of sampling in coarse-grained ice.

10) *The authors indicate that 8 areas were sampled and that at least two composite sections were made for each of the eight samples. However, in the map only 4 sampling sites are located and data is shown for only 3 samples. Why? Where are the data for SG6-B, which seems from its location in the map to sample a lower strain domain?*

Data for SG6-B are presented in figure 2. This sample was collected and analyzed prior to developing the sample preparation method for EBSD. In order to measure enough grains from the block SG6-B, the entire sample was used to create enough thin sections (7) to measure ~100 grains. Therefore we could not re-analyze it using EBSD. We present the compilation of c-axis measurements from the seven thin sections of this sample, done using a U-stage, to illustrate a particular point. While SG6-B might be from a slightly lower strain domain, there is little control of strain gradients in natural ice (see comment #3), and this sample was collected in the same intensely sheared marginal ice as SG23, SG27 and SG28. We do not expect its fabric to differ significantly from the fabrics in these. This will be clarified in the revision.

The samples we collected in the 2018 field season were concentrated along the margins and at the front of the ablation zone l. 204-205. We focused on SG23, SG27 and SG28 for the purposes of this paper because as noted, we collected more samples than the four for which we present data in this paper. These four are from a small area with well defined kinematics in the highly sheared marginal ice. The others were spread out across the glacier in various and more complex local settings, were not clustered in such a way that data could be combined to produce a CPO with a sufficient number of grains for a strong interpretation, and thus do not contribute to the arguments we present here.

11*) In l. 294, it is indicated that EBSD work is performed on 40mm x 60mm sections. However, all EBSD maps presented in the article are much smaller (25 x 25 mm on average in Fig. 6a and 3.5 x <3 mm in Fig. 7a). Why use a reduced analysis area in a study where the size of the mapped area is critical?*

The reviewer brings up a good question. The copper and aluminum ingots on which the samples were mounted were up to 40 x 60mm because that is the maximum size the SEM can analyze without significant risk of sample crashes (Prior et al 2015 show a larger sample but 40x60 is now the standard max size). This size pushes the limits of the instrument, and therefore we aimed to make sections that were not quite 60mm wide. We experimented with the width of the composite slices, initially starting with 5mm (see Fig. 7, SG23 composite 2 EBSD image—this was the first composite constructed), which did not provide many grains. We determined that for bulk CPO analysis, in order to maximize the number of grains, we needed to use more slices that were thinner. We ultimately aimed for 36 spaced slices per sample - 18 per composite - that were each approximately 2mm wide. This allowed for some extra room, which was important because different bubble concentrations throughout the sample made certain areas more fragile than others. Slices in areas with a high bubble concentration needed to be a bit wider. Ultimately, most of the composite sections were between 36 and 50mm wide. Thus it was practical considerations that limited the width of the sections we produced. This will be clarified in the revision.

For whole sections, we were interested in examining the internal structure of the largest grains, which included subgrain boundaries, and also the misorientations between grain boundaries. Many of the sections measured were mounted on the larger ingots (40 x 60 mm), but due to the limited number of these, some were mounted on smaller ingots (30 x 30 mm). All produced similar analytical results. We chose to show sections from the smaller ingots (Fig. 6a,b) because the data resolution was high (not many mis-indexed points/holes in the data, or cracks in the section) in comparison with those from the larger ingots. These sections highlight all the features we discuss.

*References:*
Azuma, N., and Azuma, K.G.: An anisotropic flow law for ice-sheet ice and its implications, Annals of Glaciol., 23, 202-208, 1996.

Gagliardini, O., Gillet-Chaulet, F., and Montagnat, M.: A review of anisotropic polar ice models: from crystal to ice-sheet flow models, Physics of Ice Core Records, 2 (68), 149-166, 2009.

Lile, R.C.: The effect of anisotropy on the creep of polycrystalline ice, J. Glaciol., 21, 475-483, 1978.

Llorens, M.-G., Griera, A., Bons, P. D., Lebensohn, R. A., Evans, L. A., Jansen, D., and Weikusat, I.: Full-field predictions of ice dynamic recrystallization under simple shear conditions, Earth Planet. Sc. Lett., 450, 233–242, 2016a.

Llorens, M.-G., Griera, A., Steinbach, F., Bons, P. D., Gomez-Rivas, E., Jansen, D., Roessiger, J., Lebensohn, R. A., and Weikusat, I.: Dynamic recrystallization during deformation of polycrystalline ice: insights from numerical simulations, Philos. T. Roy. Soc. A, 375, 20150346, https://doi.org/10.1098/rsta.2015.0346, 2017.

Pimienta, P., and Duval, P.: Mechanical behaviour of anisotropic polar ice, The Physical Basis of Ice Sheet Modeling, 57-66, 1987.

---

## Author Response (AR1)

[revised manuscript text omitted]

Response to Referee Comments

We are grateful for the referees in pointing out where more information should be provided and where clarification is needed. In the responses below, we include additional information to bolster claims made in the manuscript. We disagree with the referees on a number of points, as indicated. We have revised the manuscript accordingly, and all revisions are marked with red text.

These responses are provided in the order in which the referees' comments were made. The referee comments are italicized.

Referee No. 1

1) *"The scientific conclusions provided in the paper are not new and far too weak for the expectations of a standard research paper in The Cryosphere".*

We dispute this criticism, although accept that we may not have made the strongest case for our arguments. The method we describe is new, as we believe are the conclusions drawn from its application. We also disagree strongly with the notion that multimaxima fabrics are "not enigmatic." They are not well understood, and there are thus good reasons for focusing attention on them, as explained in point #13 below. The fact that the fabrics remain enigmatic was a prime reason for undertaking the study.

The main conclusions of the study are:

1.  This is the first study that allows a-axes as well as c-axes to be measured in coarse-grained ice by application of EBSD, and does so in a setting that is clearly dominated by shear. We have emphasized this in the abstract (l. 23-25), and in the conclusion (l. 600-601). The only other study that measured a-axes in coarse-grained ice we are aware of used the less accurate, and lower angular resolution etch-pit technique (Matsuda and Wakahama, 1978) and did not consider the fabric in relation to the kinematic setting in the ice body. [We should note that a-axes have also been measured in fine-grained Antarctic ice using both semi-automated Laue X-ray diffraction, (Weikusat et al., 2011), and EBSD (Obbard et al., 2006; Obbard and Baker, 2007; Weikusat et al., 2017). They have also been measured in sea ice using EBSD (Wongpan et al, 2018)]. These methods have all been noted, and all authors have been cited (l. 229-234). We show that a-axes are preferentially aligned and thus, in this case, slip is not isotropic in the basal plane as is often assumed, although this assumption is shown not to hold true based on recent experimental work (Journaux et al., 2019; Qi et al., 2019). Theoretically, slip is isotropic when n = 1 or 3 in the flow law, but not when n = 2 or 4 (Kamb, 1961). We have stated this in l. 77-79. To an extent yet to be explained, a-axis orientations may provide information on kinematics and rheology, and full crystallographic orientations, which can help better characterize deformation, recovery and recrystallization mechanisms (Prior et al., 2015), are therefore important.
2.  The study draws attention to the issue of interlocking grains of complex shape and the problem of individual grains being counted multiple times and thus contributing to false maxima in fabric diagrams. This problem is not new and was clearly recognized by early workers. However, little to no attention has been given to it in recent work. We now emphasize throughout the paper (including the abstract l. 27, and in greater detail in the introduction l. 171-190) that other studies have recognized and addressed this problem, although, we note that it has not been given much attention in recent literature.
3.  We provide data from separated parts of single samples and combined data from well separated samples in the same part of the glacier with fairly well-defined kinematics to suggest that what might be taken in individual samples as fabrics with three or four c-axis maxima become a simpler fabric that is very similar to what is expected conceptually for deformation dominated by slip on the basal plane under simple shear combined with some shortening normal to the shear plane. This is a new interpretation, and is demonstrated throughout the paper (see fig. 8; l. 30-35, 438-445, 494-495, 563-566).

4. The fabric pattern (both c-axis and a-axis) is similar to what is found in experiments involving simple shear, in which sub-sampling produces fabrics that could be considered individually as multimaxima fabrics. This is detailed in the discussion, section 6.3, l. 502-593.
5. We bolster microfabric evidence given by others for dynamic recrystallization involving grain-boundary migration in coarse-grained ice in contributing to the development of the crystallographic fabric (l. 472-474).

*2) "The method described, although sounding interesting, is not compared to any other type of measurement, for instance, many sample analyses over a continuous part of a core, or of a block of ice in order to provide enough grains for a good statistics".*

Comparing our method with others of handling large grain sizes is an omission in the manuscript that has now been addressed in l. 182-186. We did not claim to be the first to combine orientation data from multiple sections to overcome the problem of sampling when dealing with very large grain sizes (see Rigsby, 1951; Kamb, 1959; Gow and Williamson, 1976; Thwaites et al., 1984), nor in recognizing (addressed in l. 186-188) that individual grains may appear multiple times in a single thin section (or in separate thin sections from the same block or core segment of ice) and thus be responsible for enhancing maxima in pole diagrams (l. 188-190) (see Bader, 1951; Rigsby, 1951; Kamb, 1959; Jonsson, 1970). Our method provides a way of dealing with the specific technical challenges of using EBSD for coarse-grained ice. The time/ resource limitation for EBSD is time on the instrument and with fast EBSD speeds, the sample exchange becomes the limit. Making a composite means that we collect data equivalent to 10 to 20 full sample sections with only one exchange of samples: that means half a day of SEM time rather than 2 weeks of SEM time. This has been clarified in l. 305-309. As the above references document, the issues associated with sampling coarse-grained ice with interlocking texture were most clearly addressed in the early work on ice fabrics using the U-stage, by spacing successive thin sections between 5 and 15cm. In recent work where fabric has been analyzed using AITA or EBSD, however, the issue of the same grain possibly contributing multiple points to a fabric diagram does not appear to be addressed. For example, as outlined in l. 186-190, we could not find a description of how sampling was handled for coarse-grained ice (multiple samples from continuous core according to the referee) in Dahl-Jensen et al. (2013) (and associated supplemental data) and Montagnat et al. (2014), nor is it clear from these papers if efforts were made to consider the likelihood of multiple points contributing to the CPO from individual parent grains appearing several times in a thin section. Generally, in the more recent papers that use AITA, where multi-maxima CPOs are presented, the associated discussion is brief and simply makes note of the complex, large interlocking crystals (Dahl-Jensen et al., 2013; Fitzpatrick et al., 2014; Montagnat et al., 2014). There is no attention given to the problem of multiple sampling of single grains and thus the potential for false maxima or reasons given for the multi-maxima pattern (l. 186-190). We note that representing data using all-pixel orientations does take into account the issue of parent grains with satellite island grains, but this is only if the sample is large enough to contain a sufficient number of grains to provide a truly representative fabric. If the sample does not contain a representative number of grains, as is often the case with coarse-grained ice, then using one-point-per-grain provides a more representative fabric (fig. R1). This has been clarified in the manuscript in l. 379-384), and is addressed in detail in the appendix.

[Figure]

**Figure R1:** EBSD map and associated CPO plots for sample PIL36 deformed in uniaxial compression at -9.8°C from Qi et al. (2017), highlighting the difference in representing data as all-pixel orientations vs. one-point-per-grain orientations as a function of sample size. (a) EBSD map, with boxes representing subsampled areas. (c) CPO plots (c- and a-axes) of all pixel orientations from the entire sample area, ¼ the sample area and 1/16 the sample area. (d) CPO plots (c- and a-axes) of one-point-per-grain orientations from the entire sample area, ¼ the sample area and 1/16 the sample area. This figure was added in appendix A.

The issue of statistics is not straightforward for coarse-grained ice with the existence of multiple maxima. Any way of attempting to eliminate the effects of multiple counting of individual grains that appear more than once in a thin section or in multiple sections intersecting a single crystal would be ad hoc. Doing statistical tests while ignoring this phenomenon is of little use. Kamb's (1959) method of contouring provides a way of establishing the statistical significance of maxima in a fabric, but this is only meaningful if multiple points from the same grain are excluded. We note that use of eigenvalue methods and associated statistics is inappropriate for multiple maxima fabrics. This is now included in the appendix and in l. 133.

*3) "are we sure not to measure several times the same large crystal coming from the depth of the block, since some crystals are more than 90mm large? Owing to the fact that exact shape and location of grains are lost, there is no way to verify such a situation, as is done in figure 2 for instance. The introduction pretends that the use of a-axes measurements could provide supplementary information to check the belonging of measured areas to one single crystal or several, but this procedure is not used neither described later."*

First, it is important to note that even with the maximum size thin section (using any method of analysis), the exact shape and extent of individual grains remain unknown. We clarified this in l. 176-177. Bader (1951) and Rigsby (1968) were the first to illustrate the likely complexity of individual grains crossing successive thin sections, and we note this in l. 179-180.

Figure 2 in the manuscript is provided precisely to give an illustration of the problem within a single 2D section as shown under cross-polarized light. For this figure, we highlighted grains we *believed* to be islands of the same parent grains based on the c-axis orientations. By examining the one thin section, or even successive thin sections depending the potential size and shape of individual grains, it is not possible to state with certainty that two grains with the same or very similar c-axis orientation are branches of the same grain. One would have to trace every measured grain through an undetermined number of successive thin sections in order to say with certainty which grains are repeated in individual thin sections (clarified in l. 225-228). It is possible to confirm a common parent to island grains in a single thin section using both c-axes and a-axes. As stated in the introductions, we in fact show how this is done in figure 6 in the manuscript where three grains with the same c-axis orientation also have the same a-axis orientations, we did not just pretend to do it. We were able to use the a-axes as confirmation that tight clustering of points in composite samples that appear as maxima, likely come from the same grain, because the c- and a-axes are nearly identical, thus highlighting the probability of false maxima. This was clarified in section 6.2 (l. 486-488) by stating that the c-axis clusters are coupled with corresponding a-axis clusters, which as for the whole sections, likely indicates repeated representations of the same grain.

*4) "There is only a very weak discussion about the orientation error produced by this multi-slicing technique, although it could be quite strong, and add on at each slicing step."*

Greater attention should have been given to this important point, which we do here in order to justify the conclusions we reached concerning the fabric diagrams. We consider the process in several stages. Each sample is first squared into a rectangular prism, with one side vertical and another parallel to foliation, using guides to ensure perpendicularity. Guides are then used for each of steps 1-4 (Fig. 5), cutting the sample progressively into slabs, rods, cubes and slices. The errors involved in each stage of this process are estimated to be less than 0.5°. The error involved in slight twisting between slices during assembly into a composite section is estimated to be no more than 1°. Combining data from two or three composite sections in a sample adds only possible errors of misalignment in mounting for EBSD measurement. This is estimated to be no more than 0.5°. The largest source of error is in combining data from the three samples (Fig. 8). The reference frame for this is the foliation plane (xy-plane with vertical, x, recorded on each block when removed from the glacier.) The error in combining data from the three samples is estimated to be no more than 1°. Adding these sources of error, we estimate the uncertainties in positioning points on the pole diagrams in Fig. 7 to be no more than 4° and in Fig. 8 to be no more than 5°. The overall effects of such errors are likely to modestly diffuse rather than strengthen the maxima shown, but they will not modify the basic pattern. We assert that the measurements we have made are sufficient to establish the main features of the fabric in Fig. 8. We have detailed the error associated with the sample preparation technique (l. 323-331 and 446-453).

*5) "The too limited number of measured crystals is attributed solely to the measurement technique (AITA or EBSD) in using too small samples. This is not so true since it is possible to measure several contiguous samples*

*from an ice core – see Dahl-Jensen et al. (2013) and Montagnat et al. (2014), for instance where analyses along contiguous samples from 1 m long cores were done.”*

       We do not make such a claim about measurement techniques (see #2 above), although we did not discuss other
ways in which authors have addressed the problem. This is something that needs to be added. Early workers had
       addressed the problem of sample size by making multiple sections from different parts of a sample or core,
       spacing thin sections between 5 and 15cm intervals, (Rigsby, 1951; Gow and Williamson, 1976; Thwaites et al.,
       1984) or from more than one sample (Kamb, 1959). We have addressed this in l. 182-186.

*6) “One of the main conclusions is related to the observation, in some previous studies, of multi-maxima, and
       their attribution to a too limited number of crystals. This result is not so new and was intuited by most of the
       authors responsible for the mentioned studies. Experimental observations such as the ones from Qi et al. 2019
       shown in the paper enabled to confirm this intuition already since, with a larger number of grains, the
       multimaxima texture do not exist anymore.”*

Our reanalysis of the Qi et al. (2019) dataset is a new and unique contribution. Using their data, we emphasize
       that if you sample a small subset (a statistically insufficient) number of grains, one can produce an apparent
       multi-maxima CPO from an otherwise well-defined simple shear CPO.  This is not a point made by Qi et al.
       (2019), who were not concerned with multimaxima fabrics and who did not consider subsets of their data. We
       believe this was well stated, and ask the reviewer to please refer to section 6.3 starting on line 597.
       7) *“Although there is one dominating orientation in the combination of samples presented on figure 8 the multi-
       maxima remains, with 3 main orientations. So the result is not so obvious and can not lead to such a firm
       conclusion”.*

We admit that the pattern is not clear cut, but the strong split maximum lying in the plane perpendicular to the
       foliation and containing the vorticity axis is just what is expected for simple shear, as observed in in torsion
       experiments (See paper, lines 523-537), and accentuated by adding a component of compression normal to the
       shear plane (fig. R2). One of the two weak submaxima in the plane normal to foliation and parallel to the shear
       direction (fig. R2) is what is expected in simple shear, as in the experiments of Qi et al. (2019), which also show
a hint of a submaximum offset in the opposite direction, a second sub-maximum (fig. R2) that is more apparent
       in our samples. We believe this is clearly discussed in the manuscript, but have modified fig. 10b in the paper
       using fig. R2, and the associated figure caption in l. 1086-1088.

[Figure]

**Figure R2:** Schematic pole figures highlighting the relationship of the CPO to the foliation/shear plane (gray), the vorticity
axis (red) and the plane normal to the shear direction (navy blue).

8) *“Moreover, to be more affirmative, one would have needed more results, on various samples which is not
       shown in this study.”*

       Yes, of course, more data on more samples is desirable, but the resources available to conduct this work were
       limited. We consider this a self-contained study that should prompt further research. We might stress the amount of work involved in extending such an enterprise. Each individual large sample takes hours of physical labor to collect, insulate and haul over rough terrain to a freezer at the research station. Then preparing the samples for transport in several stages to Otago in New Zealand, while ensuring that they stay well below freezing at all times, requires careful packing in large coolers, each of which can only contain 4 samples, and close monitoring during transit. Additionally, there are particular challenges of working with coarse-grained ice using EBSD.

Each large section takes >1 hour to analyze at a coarse step size (50µm), additional time to analyze any areas of interest in finer detail, and another hour to do a sample exchange, run the sublimation cycle to clean frost off of the sample for imaging, bring the stage down to the correct temperature, and set up another analysis. *If* all goes smoothly, only 3-4 sections can be analyzed per day. We have added these details (l. 367-371). Beyond this, there is the time taken to prepare the composite sections, steps which are laid out in l. 310-322 (fig. 5) and take

~5-6 hours for each sample, and an additional ~2 hours to prepare whole sections once the slabs of each sample are polished, allowed to sublimate overnight and photographed.

*9) "The other conclusion related to grain boundary migration dominating dynamic recrystallization processes in the studied conditions is not new at all, and simply confirm the observations by most authors working on*

*dynamic recrystallization mechanisms in warm conditions see for instance De la Chapelle et al., 1998, but also the laboratory work by Jacka and co-authors, or the most recent work by Journaux et al. 2019."*

Yes, many studies attribute multimaxima fabrics to dynamic recrystallization dominated by grain boundary migration, citing the large interlocking nature of the grains that form at high temperatures (Rigsby, 1955; Gow and Williamson, 1976; Gow et al., 1997; Duval, 2000; Diprinzio et al., 2005; Gow and Meese, 2007; Montagnat et al., 2014). This is not of course new, nor did we claim it to be, though we acknowledge it more clearly in l. 210-212. However, we do provide additional textural observations in support of this assertion (l. 471-473): individual grains lack significant internal distortion, and no visible shape preferred orientation. We could add evidence of grain boundary drag around bubbles (e.g. Fig. 6a), similar to pinning effects discussed by Evans et al. (2001).

*10) "Although the authors used the review paper by Faria and co-authors (2014), it is necessary to provide the references of the original works to whom the credit should be give. Otherwise the community will little by little lose track of these original work and the credit will only go to the one who wrote the review."*

We agree with this comment. The point has been addressed in l. 57, 73-74, and 143-145.

*11) "About open cones CPO in polar bore holes, once again the citation of Faria et al. 2014 is inappropriate, since Faria and co-authors did not make any measurement along deep ice cores, and this is not true I think, that*

*this type of CPO is not observed along polar ice cores."*

We also agree with the second part of this statement. There are certainly fabrics close to open cones (sometimes called small circle girdles) in the upper parts of polar ice cores (e.g. Ross ice shelf, Gow and Williamson, 1976; Byrd Station, Gow and Williamson, 1976; Camp Century; Herron and Langway, 1982; Cape Folger, Thwaites et al., 1984; Dye 3, Herron et al., 1985; Siple Dome, DiPrinzio et al., 2005; Siple Dome, Gow and Meese, 2007; NEEM, Montagnat et al., 2014). In most of these cases these fabrics are identified as such. This has been clarified (l. 124-134).

*12) "Also less clear than in experimental work, some CPO very close to open cones are observed in the bottom*

*od the GRIP, GISP2, BYRD cores for instance."*

We agree with this, too. Our statement on this needs clarification. Sometimes CPOs at the base of ice sheets are identified as possible open cones/ small circle girdles or modifications of open cones/small circle girdles (e.g. Byrd Station, Gow and Williamson, 1976; Tison et al., 1994; GRIP, Thorsteinsson et al., 1997; GISP2, Gow et al., 1997; Siple Dome, DiPrinzio et al., 2005; Siple Dome, Gow and Meese, 2007), even though these types of fabrics typically show some clustering that is interpreted as a multimaxima CPO. It is important to note that the eigenvalue technique of fabric representation, often used with AITA analyses, does not distinguish between small circle girdles and multimaxima fabrics (Fitzpatrick et al., 2014). This is coupled with our response #11 and thus has been clarified (l. 124-134).

*13) "Line 118: the multi-maxima CPO is not enigmatic and some hypotheses were given by different authors… See for instance De la Chapelle et al. 1998"*

We disagree strongly with the statement that multi-maxima CPO are not enigmatic. Indeed several hypotheses
have been advanced to explain these fabrics, but there is no consensus on their significance. We address here at
some length the reasons for our disagreement. Our explanations, which are detailed below, are interspresed
throughout section 2, l. 171-217.

Multi-maxima crystallographic fabrics have been recognized since the 1950s, though there has been little focus
on them in recent years, perhaps due to the large grain size and sampling difficulties. They have been
documented in coarse-grained ice in many studies of valley glaciers (Rigsby, 1951; Meier et al., 1954; Kamb,
1959; Higashi, 1967; Jonsson, 1970; Fabre, 1973; Vallon et al., 1976; Tison and Hubbard, 2000; Hellmann et
al., in review), in deep warm parts of polar ice sheets (Gow and Williamson, 1976; Matsuda and Wakahama,
1978; Russell-Head and Budd, 1979; Gow et al., 1997; Diprinzio et al., 2005; Gow and Meese, 2007;
Montagnat, 2014; Fitzpatrick et al., 2017; Li et al., 2017), and at the margins of polar outlet glaciers (Kizaki,
1969). They have also been produced in experiment, in torsion tests (Steinemann. 1958) or torsion combined
with compression (Duval, 1981; Russell-Head, 1985) and in compression alone combined with annealing
(Maohan et al., 1985). Despite this attention, we believe these fabrics remain poorly understood,

This is why we believe this to be the case.

**First**, there is the question of whether or not the maxima are truly distinct.  This is due to the possibility of there
being many island grains of a single parent creating false maxima (Rigsby, 1951; Kamb, 1959) or there being an
insufficient number grains to define the fabric (Kamb, 1959).  Tests of significance do not account for repeated
counts of island grains. The most likely confusion is between true multimaxima fabrics and small-circle girdle
distributions (Kamb, 1972; Maohan et al., 1985; Thwaites et al., 1984).  We note that per-pixel fabric diagrams
produced whether by AITA or EBSD automatically bias towards large grains and can potentially produce
spurious maxima (fig. R1). Plotting grains as one-point-per-grain for bulk CPO analyses can reduce this bias.

**Second,** and likely because there may be single grains with island satellites producing their own individual
maxima, the number of maxima recorded is variable, from three to five or six (e.g. Rigsby, 1951, 1960; Kizaki,
1969; Jonsson, 1970), although four is the "ideal" number arranged in a rhomboid or diamond pattern (Rigsby,
1951, 1960); however the shape of the pattern is variable. The angles between the maxima and the "center of
gravity" of the individual clusters vary between 25 and 45 degrees (Rigsby, 1951; Kamb 1954; Jonsson, 1970;
Gow and Williamson, 1976; Russell-Head, 1985; Budd and Jacka, 1989).

**Third** is the relationship of the fabrics to the state of stress or strain.  For fabrics that are most distinctly of four-
maxima type, it is commonly assumed that the fabric is related to the state of stress and that the maxima reflect
basal planes aligned in orientations of high shear stress (Duval, 1981), even though there are only two planes of
maximum shear stress in a general (triaxial) state of stress. It has been suggested that these fabrics develop in ice
that has undergone prolonged shear (Kamb, 1959, Higashi, 1967), and also that they may represent partial
annealing in ice that may be under a low state of stress (Higachi, 1967; Budd and Jacka, 1989) or nearly
stagnant conditions (Russell-Head and Budd, 1979). There is thus uncertainty about the stress level under which
the fabrics develop and how much of the history of deformation experienced by the ice is reflected in the fabric.
Many authors have noted that the center of the set of maxima lies near the pole to foliation (Rigsby, 1951; Meier
et al., 1954; Kamb, 1959, Kizaki, 1969; Jonsson, 1970), which in marginal ice is the plane of high shear stress
and also one of high shear strain. However, similar fabrics are found near the center of glaciers where shear
parallel to foliation is a minimum, yet maxima are still centered around the foliation pole (Fig. R3; e.g. Kamb,
1972, Fig. 17b) or parallel to the direction of maximum compression (Hellmann et al., in review). If the fabric
reflects the state of ambient stress, with no memory of stress or strain history, there should be a consistent
relationship between the fabric elements and principal stress directions and there should be no distinction
between fabrics developed under coaxial and non-coaxial kinematics.  This does not appear to be the case
because in simple shear the maximum principal compressive stress is at 45° to the shear plane and the maxima
are arranged about the normal to the shear plane (as is found along the margins and at the base of valley
glaciers), whereas in coaxial deformation, as would be expected in the near-surface central parts of valley
glaciers, the maxima are symmetrically arranged about the maximum compression direction (fig. R3; Hellmann
et al., in review) or the pole to foliation (fig R3; Kamb, 1972, Fig. 17b).  Thus there is ambiguity about the
relationship between the maxima and the orientation of principal stresses (as inferred from strain rates or from
modeling) and the relationship between the maxima and orientation of the principal directions of cumulative
strain, whether the strain history is one of coaxial or non-coaxial type.

[Figure]

**Figure R3:** (a) Schematic relationship between classic four maxima fabric pattern and inferred state of stress in valley glaciers. (b) pure shear, found near the surface in the center of glaciers in the ablation zone where ice is in longitudinal compression, with $s_1$ is horizontal and perpendicular to foliation, which is typically nearly vertical and transverse to glacier flow. (c) simple shear, with $s_1$ inclined at 45° to the foliation, which is steep and parallel to the valley sides.

**Fourth,** in addition to these uncertainties, it has been suggested by (Matsuda and Wakahama, 1976) that the maxima represent crystals that are in a mechanical twin relationship with one another. Duval (1981) suggests the possibility of annealing twins rather than mechanical twins. The texture in thin section gives little indication of twinning, however. The possibility of twinning can only be investigated if both c- and a-axes are known.

14) *"Part 6.3: lines 430 to 326, the dynamic recrystallization processes are mentioned as a likely difference between the experimental and natural conditions, owing to the difference in strain rate. Although already in the experimental conditions is dynamic recrystallization very active, especially at this high temperature, and the driving force for GBM is even stronger since it is associated to the storage of dislocations at GB, the latter being expected to be stronger at high relative strain rate. At lower strain rate, we expect the dislocation storage to be slower relative to GB mobility."*

There are data to support our statement.
Cross and Skemer (2019) using empirical data show that dynamic recrystallization is fastest under high temperature, low stress conditions, although also stating that this conclusion needs testing because it is counterintuitive. In any case, both grain boundary mobility (function of temperature) and driving force (function of the storage of dislocations as a result of stress) are important and the scaling between these two from experiment to natural conditions is not known. We have clarified these statements l. 584-591.

*may question if the technique proposed is really effective (the spacing used for the sectioning is probably still*
*smaller than the maximum dimension of the grains). One may therefore question why should one prefer this*
*method to the even more traditional one (at least in geology) of collecting oriented samples in a series of*
*profiles normal to the shear zone trend and then add up the data for samples with similar positions across the*
*shear zone. This second approach would allow to: (1) spread the sampling a much larger volume, (2) preserve*
*the relation between CPO and microstructure, which is essential for discussing the role of deformation and*
*recrystallization processes on the evolution of the CPO, and (3) collect data for variable finite strains (which is*
*missing here and would have been extremely useful to discuss some features, such as the deviation of the [0001]*
*maxima relatively to the normal to the shear plane along the plane normal to the shear direction or how the*
*CPO evolves with finite strain).*

Doing what the referee proposes would be impractical for the situation encountered in this and most valley
glaciers. We do not have clear markers of shear strain that allow us to document a strain gradient across the
marginal ice, but we can reasonably assume that fairly closely spaced samples come from a homogeneously
deformed volume of ice. Collecting, handling, transporting and preparing for analysis many more large samples
was beyond the resources available to us. We could not do on a large scale what Hudleston (1977) was able to
do on the scale of a single thin section (and what the referee suggests here) for a small-scale shear zone in cold
ice at the margin of the Barnes Ice Cap. However, more systematic sampling, allowing for individual slices of
each composite to be spaced by >15cm would be beneficial and appropriate for a follow-up study. We have
expressed the need for future studies that undertake a more systematic sampling approach in the conclusion l.
608-610.

3) *In conclusion, neither the results nor the technique are completely new. If the article is to be published (I do*
*not know the journal well enough to make a recommendation), it has to be revised to present in a more objective*
*way its actual contribution: new data on the evolution of CPO of ice in natural shear zones, which confirm the*
*current knowledge on the subject: simple shear under high homologous temperature produces a CPO*
*characterized by concentration of [0001] axes normal to the shear plane.*

We have attempted to highlight and clarify what is new and unique to our study. We refer to the response #1 of
the first referee to emphasize what is new in our study. We should note that all natural ice deformation is under
conditions of high homologous temperatures. There is almost no new data for the evolution of CPO of natural
ice in shear zones, because there is very little close control of strain gradients in natural ice. Nearly all the
published data comes from laboratory experiments. As far as we are aware there is still only one study of
fabrics in natural ice constrained to be from a well-defined shear zone (Hudleston, 1977). We have added this
for clarification in section 2 l. 145-149.

4) *Moreover, the discussion should be reinforced and present a comparison of the observations with all*
*available experimental data in simple shear (why focus the comparison on a single set of experiments?)*

As far as we are aware there are only two published sets of experiments that document both c-axis and a-axis
fabrics in simple shear in ice, and those are the ones by Qi et al. (2019) and Journaux et al. (2019), both of
which we cite and the results of which we compare with our data. We have clarified this in section 6.3 l. 499-
500. In the case of Qi et al. (2019), we show how taking a subset of the data leads to less well defined fabric
patterns that might be compared to natural fabric patterns with limited grain counts. We do cite other
experiments done in simple shear or in simple shear plus compression normal to the shear plane, but the data in
these is not presented in way that allows for direct comparison with our data or the data of Qi et al. (2019).

*5) The rather 'surprising' observations of: (1) lack of a maximum of <a>-axes parallel to the flow direction and (2) the deviation of the [0001] maxima relatively to the normal to the shear plane along the plane normal to the shear direction - should be discussed in a more effective way. The present discussion, although long, does not propose any clear explanation for neither of the two observations.*

We do not have a good explanation for the first point here, and the switch with increasing strain from a-axes perpendicular to flow at low strain to parallel to flow at high strain was not explained by Qi et al. (2019) or Journaux et al. (2019) in their experiments.  The second point we do discuss (in the original manuscript, l. 379-385) though we have modified the manuscript to include a more effective discussion (l. 523-537) as detailed below. The deviation or spreading of the main [0001] maximum in a plane normal to the shear plane and in a
direction perpendicular to the shear direction (see response #1, fig. R2) is found both in simple shear experiments (Kamb, 1972; Bouchez and Duval, 1982; Journaux et al., 2019) and in experiments involving simple shear with the added effect of compression or flattening normal to the flow plane (Kamb, 1972; Budd et al., 2013; Li et al., 2000). The combination of uniaxial compression (cone distribution about the compression axis) with simple shear (single maximum perpendicular to the shear plane for large strains) provides the clearest
explanation for the split maximum (Kamb, 1972; Budd et al., 2013). Bouchez and Duval (1982), and Journaux et al. (2019), however, observe the tendency for the main c-axis maximum to spread in experiments using fixed plattens where compression could not be a factor. Li et al. (2000) attribute the spreading to transverse extension accompanying the flattening of the sample during deformation in their experiments. Numerical simulations by Llorens et al. (2016a, 2017) show this spreading does occur in simple shear with no flattening strain, and that it
is enhanced by dynamic recrystallization. It is most pronounced at low strain rates. Qi et al. (2019) suggest that the spreading increases with increasing shear strain. In our case, at the margins of Storglaciären, the ice is deforming at high temperatures, low strain rates, and to high strain, consistent with conditions that enhance spreading in experiments and in modeling.

*6) The statements presenting the relation between microphysical processes and CPO evolution in the abstract, introduction, discussion, and conclusion lack precision and give the (false, in my point of view) impression that CPO evolution is mainly controlled by recrystallization (cf. lines 15 & 58-60) or that dynamic recrystallization may completely reset the CPO (cf. lines 30-32 & 434-436). As I see CPO is produced by dislocation glide and recrystallization modifies it, by creating new orientations (most often only dispersion around the orientation of*
*the parent grains) and selectively consuming others when grain growth is effective as it is the case here. The first process certainly buffers the increase in the CPO intensity, but not fully resets the CPO. The second may significantly change the CPO when grain growth is orientation dependent*

We agree entirely with how the referee interprets the CPO and thought that is what we stated in the manuscript. We apparently have given a false impression. We have attempted to clarify by modifying statement in lines 18-19, 66, 36-38 and 588-591.

*7) Which are the arguments which justify that low strain rates should enhance dynamic recrystallization and grain growth (l. 434)? I would rather propose the opposite as the forces associated with dislocation density*
*gradients should be smaller at low strain rates.*

  See the response to point #14 of the first referee. We have modified and expanded on the text in lines 583-591.

*8) Referencing is often loose and there are many places where pertinent references are missing. For instance, l. 61, Wenk and Christie (1991) is not the best reference in a phrase dealing with CPO-induced mechanical anisotropy when there are a large number of studies that investigated precisely this effect (cf. review by Gagliardini et al. 2009 and references therein).*

This is a fair point, which can be addressed, although we believe that Wenk and Christie (1991) is an important reference as these authors discuss the effect of CPO on the internal flow strength of rocks (relating back to the many important purposes for studying ice l. 42-45). Examples of additional references, relating specifically to CPO development modifying the internal flow strength of polycrystalline ice include: Steinemann, 1958; Lile, 1978; Pimienta and Duval, 1987; Alley, 1988; Alley, 1992; Azuma and Azuma, 1996; Gagliardini, 2009. We have added these references in l. 69-71. We have also tightened other loose referencing as indicated in response
to reviewer #1.

  *9) The aims of the article should also be redefined. Those stated in l.79-82 were probably the initial aims of the study, but given the results, they cannot be the aims of the article.*

We are puzzled by this comment. The aims of the study were given in the original manuscript in lines 79-82. The only thing we might change is to replace the word "fully" by "better," since we have not fully addressed the issue of sampling in coarse-grained ice. We have done this (l. 94-97).

10) *The authors indicate that 8 areas were sampled and that at least two composite sections were made for each of the eight samples. However, in the map only 4 sampling sites are located and data is shown for only 3 samples. Why? Where are the data for SG6-B, which seems from its location in the map to sample a lower strain domain?*

Data for SG6-B are presented in figure 2. This sample was collected and analyzed prior to developing the sample preparation method for EBSD. In order to measure enough grains from the block SG6-B, the entire sample was used to create enough thin sections (7) to measure ~100 grains. Therefore we could not re-analyze it using EBSD. We present the compilation of c-axis measurements from the seven thin sections of this sample, done using a U-stage, to illustrate a particular point. While SG6-B might be from a slightly lower strain domain, there is little control of strain gradients in natural ice (see comment #3), and this sample was collected in the
same intensely sheared marginal ice as SG23, SG27 and SG28. We do not expect its fabric to differ significantly from the fabrics in these. We have added a sentence to the figure 2 caption (l. 1003-1005) for clarity.

     The samples we collected in the 2018 field season were concentrated along the margins and at the front of the ablation zone (original submission, l. 204-205). We focused on SG23, SG27 and SG28 for the purposes of this
paper because as noted, we collected more samples than the four for which we present data in this paper. These four are from a small area with well defined kinematics in the highly sheared marginal ice. The others were spread out across the glacier in various and more complex local settings, were not clustered in such a way that data could be combined to produce a CPO with a sufficient number of grains for a strong interpretation, and thus do not contribute to the arguments we present here. We clarified this in section 4.1 l. 283-285.

     11) *In l. 294, it is indicated that EBSD work is performed on 40mm x 60mm sections. However, all EBSD maps presented in the article are much smaller (25 x 25 mm on average in Fig. 6a and 3.5 x <3 mm in Fig. 7a). Why use a reduced analysis area in a study where the size of the mapped area is critical?*

The reviewer brings up a good question. The copper and aluminum ingots on which the samples were mounted were up to 40 x 60mm because that is the maximum size the SEM can analyze without significant risk of sample crashes (Prior et al 2015 show a larger sample but 40x60 is now the standard max size). This size pushes the limits of the instrument, and therefore we aimed to make sections that were not quite 60mm wide. We experimented with the width of the composite slices, initially starting with 5mm (see Fig. 7, SG23 composite 2
EBSD image—this was the first composite constructed), which did not provide many grains. We determined that for bulk CPO analysis, in order to maximize the number of grains, we needed to use more slices that were thinner. We ultimately aimed for 36 spaced slices per sample - 18 per composite - that were each approximately 2mm wide. This allowed for some extra room, which was important because different bubble concentrations throughout the sample made certain areas more fragile than others. Slices in areas with a high bubble
concentration needed to be a bit wider. Ultimately, most of the composite sections were between 36 and 50mm wide. Thus it was practical considerations that limited the width of the sections we produced. We have added this information in section 4.2 l. 338-349.

     For whole sections, we were interested in examining the internal structure of the largest grains, which included
subgrain boundaries, and also the misorientations between grain boundaries. Many of the sections measured were mounted on the larger ingots (40 x 60 mm), but due to the limited number of these, some were mounted on smaller ingots (30 x 30 mm). All produced similar analytical results. We chose to show sections from the smaller ingots (Fig. 6a,b) because the data resolution was high (not many mis-indexed points/holes in the data, or cracks in the section) in comparison with those from the larger ingots. These sections highlight all the
features we discuss. We have added this information in section 4.2 l. 349-353 and section 5.2.1 l. 415-418.

---

## Author Response (AR2)

[revised manuscript text omitted]

Response to Referee Comments

We thank the referee for her comments on the revised manuscript. We respond to these comments below, and have revised the manuscript accordingly. All revisions to the manuscript are marked with red text.

The referee comments are italicized.

*The statement in the conclusion and discussion that at low strain rates, dynamic recrystallization should be more effective deserves, however, some clarification. Dynamic recrystallization depends on the work rate, which is usually higher at high strain rates, and on finite strain. So I would guess that the reason for stronger effect of dynamic recrystallization on the CPO evolution in the studied natural setting is rather the higher finite strain.*

We should expect, as the referee notes, dynamic recrystallization to be greater in the natural case than in the experiments, since the deformation proceeded to higher strains in nature than in the experiments. Yet the a-axis CPO suggests a greater similarity between the low-strain experiments and nature than between the high strain experiments and nature, suggesting a weaker effect of recrystallization on the CPO in nature than in the experiments, not stronger. Hirth and Tullis (1992) is cited to support the claim that dynamic recrystallization is effective at low stresses and high temperatures, and Cross and Skemer (2019) to suggest that it may be more effective under these conditions than at high stresses and temperatures. With the high finite strain experienced by our samples the ice must be completely recrystallized, with further strain producing further recrystallization. The idea here is that intense recrystallization in nature may continuously modify the fabric so that it does not attain the degree of development found in the experiments. In any case, it is certainly the case that dynamic recrystallization is highly effective in nature under low stress, high temperature conditions. We have clarified this in the discussion l. 574-593.

*Similarly [to the previous comment], for the statement in l. 583 "Dynamic recrystallization and grain growth are enhanced at low strain rates". Dynamic recrystallization involves two processes, relative grain growth rates (relative to deformation rates) are higher at lower strain rates and higher temperatures, allowing for coarser recrystallized grains. However, absolute recrystallization rates are not faster.*

We agree with this statement and it would be best not to state that recrystallization rates are necessarily faster under lower strain rates and higher temperatures, although the data of Cross and Skemer suggest this may in fact be the case. We have replaced "enhanced" with "effective."

*A point that was already highlighted in the first reviews is the comparison with experimental results: The authors acknowledge that there are two datasets, but compare their data only to those obtained by their group. Why? This gives the impression that the present data is only consistent with this one, which is not true since the general features of the two pre-existing datasets are coherent.*

The results of these two sets of experiments are coherent, exhibiting two clusters of c-axes, a strong cluster normal to the imposed shear plane at all strains, and a secondary cluster in a profile plane antithetic to the imposed shear direction at lower strains. Both studies are characterized by microstructures similar to those along the margin of Storglaciären, and highlight the disappearance of the weaker maximum, and an enhancement of the stronger maximum with high shear strains. We have modified the text to emphasize the similarities between the two experimental studies so that the impression is not that our data is only consistent with one of the experimental studies. We focused our detailed comparison on the data from Qi et al., 2019 because the data sets were open access and readily available.

*Concerning line 529: if in many cases, splitting of the <c> axis distribution in two maxima aligned in the plane normal to the shear direction is observed in absence of transpression, transpression cannot be the clearest explanation for this observation. In the conclusion of this paragraph, it would be better to clearly state which are the conditions you think better explain the splitting of the c-axis maxima in the studied case. It would also be*

*interesting to discuss the processes that allow for dynamic recrystallization to produce such a change in the c-axis distribution.*

1160 As reflected in the text, the explanation for splitting the c-axis maxima is not entirely clear-cut. There are many proposed explanations, a combination of which likely tells the story along the margin of Storglaciären. We have rearranged the text such that the conclusion of the paragraph clearly states the conditions we believe are better for splitting the c-axis maximum in our studied case. Here, strain rates are low, consistent with models by Llorens (2016a, 2017), finite strains are high, consistent with Qi et al. (2019), and there is a component of

1165 compression normal to the shear plane due the confinement of the glacier between the valley walls, consistent with experiments by Kamb (1972) and Budd et al. (2013). The processes that allow for dynamic recrystallization to produce a split c-axis maximum remain unclear from experiments and theory why this splitting occurs. In our case it makes sense with the added component of shortening, but we are not sure why it should occur without that.

1170

*Minor comments from the referee*

*The statement in lines 187-188 is rather harsh. Are you really sure that none of these previous studies gave enough consideration to this important and rather first-order limitation of the observations?*

1175

Our intent is not to criticize these works, since the fabric in coarse-grained ice was not the focus of these ice core studies, but to point out that the discussion provided gave no indication that these complications in working with coarse-grained ice were addressed. We adjusted the text in l. 181-186 so that it does not come off so harshly.

1180

*Line 210, "that" is missing before "recrystallization"*
We have addressed this. L. 206

*Line 235: there are other groups that perform routine EBSD on ice*

1185

Yes, we acknowledge that other groups perform EBSD on ice. We have highlighted examples of some of these studies throughout the manuscript (e.g. l. 225-227), all of which were restricted to smaller sample sizes. The intent in this paragraph is not to give an exhaustive list of previous EBSD studies, but to highlight some of the milestones that made our work on large sample sizes possible.

1190

*Lines 440-441: Words missing: The composite pattern has one c-axis maximum ROUGHLY perpendicular to the shear plane, that is elongated or split in TWO MAXIMA ALLIGNED a plane normal to the shear direction.*

This has been corrected such that the sentence now reads: The composite pattern has one c-axis maximum

1195 roughly perpendicular to the shear plane, that is elongated or split into two maxima aligned in a plane normal to the shear direction, and an a-axis girdle parallel with the shear plane with a concentration of a-axes perpendicular to the shear direction (parallel to the inferred vorticity axis of flow). L. 436-439.

---

## Author Response (AR3)

**Response to editor O. Gagliardini**

The editor's comments are italicized.

I have nevertheless a last remark regarding plotting pole figures using orientations of pixels or grains. Just before the results section, you have a paragraph which discuss this as well as the Appendix material. I feel that your conclusion that "If the sample does not contain a representative number of grains, as is often the case with coarse-grained ice, then using one-point-per-grain provides a more representative fabric" is a bit weak and not really supported by any references or objective arguments of what should be a more representative fabric? Some years ago (Gagliardini et al., 2003), I conducted a modeling work to study the influence of accounting for the grain area when evaluating statistic parameters used to describe polycrystal fabric. The conclusion was that accounting for the grain area greatly improve the fabric description, in opposition with your current affirmation. I should have mentioned this publication before in the review process, but I am always reluctant because it is not the role of an editor to promote his own work. Nevertheless, I think in this particular case, and because they have not been that much work on that very specific subject, I would be interested to know how our findings based on synthetical fabrics (for which we know what is the real fabric) can support or not your statement about pixel versus grain orientations.

We should have been aware of your 2004 paper, and thank you for bringing it to our attention now. We have modified the text in the appendix in light of this awareness and your above comments. The modified text is highlighted in red.

The issue of using one-point-per-grain vs. all pixel orientations for CPO plots is complex. We agree that if the volume of ice sampled is fully representative, such that the results of two samples are the same, and the assumption that the measured area of a grain in cross section is the mean projected area of the grain, then measuring all pixel orientations is better statistically, as you demonstrate. In particular, the weighted area representation based on all-pixel measurements is clearly the best procedure to use for providing estimates of bulk physical properties, such as the polycrystal elasticity tensor. We have highlighted this in the manuscript appendix (1.635-636). If there are no repeat or 'island' grains, then one-point-pergrain and all pixel measurements will show the same basic pattern of CPO (See fig. A1), that is, the same fabric elements – girdle or point clusters – and these are important for identifying microstructural processes. If there are island grains, then the basic assumption of grain shape in the area-weighted method is violated and the significance of the fabric pattern by either method becomes uncertain.

If the volume of ice is not representative, which is almost certainly the case with our samples, then all- pixel representations will be biased towards large grains (or in many cases, the largest grain in the section analysed, which may not be the largest or close to largest grain in a representative volume). Several large grains may then be responsible for maxima on the fabric diagram and mask the true fabric elements that would emerge if a truly representative volume had been sampled. One-point-per-grain analyses in such instances can reduce this bias, but still may include repeated grains. This is what we show in Fig. A1 where the main fabric element (small circle girdle) is clearer in the one-point-per-grain diagram than the all-pixel diagram when the sample size is reduced. For the largest sample size, there is little difference.

In our case, we do not have a representative area on the scale of one section, nor is it likely that we have a representative volume in one block of ice, which we sample by creating composite sections in order to make EBSD practicable. By combining data from three blocks from what we believe to be a homogeneously deformed volume of ice, we can reduce the bias further and better approach a representative volume. Because of the presence of the large, ameboidal shaped grains, we believe that one-point-per-grain provides a better indication of the fabric elements than all-pixel representation.

As we have pointed out, dealing with large crystals of complex and interconnected shapes remains a problem, and the grain size and shape remain largely undefined. As a result, until we can better define the size and shape of such ice, in order to look at the most representative CPO patterns to assess microstructural processes, we believe the one-point-per grain analyses provide a better representation of the CPO.

**Full crystallographic orientation (c- and a-axes) of warm, coarse-grained ice in a shear dominated setting: a case study, Storglaciären, Sweden**

5 Morgan E. Monz1, Peter J. Hudleston1, David J. Prior2, Zachary Michels1, Sheng Fan2, Marianne Negrini2, Pat J. Langhorne2 and Chao Qi3

[revised manuscript text omitted]